



# A bootstrap method to estimate the influence of rainfall spatial uncertainty in hydrological simulations

Ang Zhang[1], Haiyun Shi[1,2,3,4], Tiejian Li[1,2,3], Xudong Fu[1]

[1]State Key Laboratory of Hydroscience and Engineering, Tsinghua University, Beijing, 100084, China
[2]State Key Laboratory of Plateau Ecology and Agriculture, Qinghai University, Xining, Qinghai, 810016, China
[3]School of Water Resources and Electric Power, Qinghai University, Xining, Qinghai, 810016, China
[4]Department of Civil Engineering, The University of Hong Kong, Hong Kong, China

*Correspondence to*: Tiejian Li (litiejian@tsinghua.edu.cn)

**Abstract.** Rainfall stations with a certain number and spatial distribution supply sampling records of rainfall processes in a
river basin. Uncertainty may be introduced when the station records are spatially interpolated for the purpose of hydrological simulations. This study adopts a bootstrap method to quantitatively estimate the uncertainty of areal rainfall estimates and its effects on hydrological simulations. The observed rainfall records are first analysed using clustering and correlation methods, and possible average basin rainfall amounts are calculated with a bootstrap method using various combinations of rainfall station subsets. Then, the uncertainty of simulated runoff, which is propagated through a hydrological model from the spatial
uncertainty of rainfall estimates, is analysed with the bootstrapped rainfall inputs. By comparing the uncertainties of rainfall and runoff, the responses of the hydrological simulation to the spatial uncertainty of rainfall are discussed. Analyses are performed for three rainfall events in the upstream of the Qingjian River basin, a sub-basin of the Yellow River. Using the Digital Yellow River Integrated Model, the results show that the uncertainty of rainfall estimates derived from rainfall station network has a direct influence on simulated runoff processes. This quantified relationship between rainfall input and
simulation performance can provide useful information on managing rainfall station density in river basins. The proposed method could be a guide to quantify an approximate range of simulated error caused by the spatial uncertainty of rainfall input.

## 1 Introduction

Hydrological models have been widely used in numerous fields such as flood forecasting, water resource estimation, water
project planning, hydrological station planning, and water quality monitoring (Beven, 2001; Cibin et al., 2014). At present, a significant goal of existing research efforts in hydrological modelling is to improve the accuracy of runoff simulation and prediction from improving the structures of the hydrological models, the calibration of models, or the accuracy of the input data. Rainfall is regarded as the key driving input of hydrological models, and the influence of its high spatial and temporal variation on the simulation accuracy cannot be ignored (McMillan et al., 2011). Currently, rainfall data from radar and
remote sensing, which can provide areal estimates, are applied in hydrology and other fields; however, rainfall data from





rainfall stations are still indispensable because the station data are regarded to be relatively accurate and reliable than the radar data at the point where a station locates (Sivapalan and Blöschl, 1998; Xu et al., 2015). Therefore, the spatial distribution of rainfall stations still plays an important role in the accuracy of basin rainfall estimates, as well as hydrological simulations.

Over the past several decades, many approaches have been developed to study the spatial variation of rainfall, and numerous mathematical formulations have been proposed to identify the errors and uncertainties from rainfall measurements, including data-driven and analytical methods (Athira and Sudheer, 2015; Mirzaei et al., 2015). An important question is what spatial-temporal scales of rainfall measurement are required for estimating rainfall with an acceptable accuracy. For example, a simple rule was proposed by Villarini et al. (2008) to determine the number of rainfall stations required to estimate areal

rainfall with a prescribed accuracy (e.g., more than 15 stations over an area of approximately 135 $km^2$ were necessary to estimate the true areal rainfall at a three-hour scale with an error less than 20%). As the spatial variability of rainfall can cause large variations in simulated runoff, the propagation of error from rainfall to runoff has also been investigated in numerous previous studies. For example, Wilson et al. (1979) indicated that the differences in simulated runoff could be quite significant when the rainfall inputs of the Fajardo basin, with an area of 26.5 $mi^2$ in Puerto Rico, were changed. Faurès

et al. (1995) proved that a single rainfall station with a uniform rainfall assumption could lead to large uncertainty in runoff estimation. Lopes (1996) examined the effect of uncertainty in spatial estimates of rainfall on the prediction of runoff volume, and the results showed that the density of the rainfall station network could greatly influence the catchment response predictions when rainfall stations were randomly excluded one by one from the analysis. Moulin et al. (2009) quantified the effect of mean areal rainfall estimation errors on runoff simulation in three small to medium-sized catchments with areas

ranging from 60 to 3200 $km^2$.

Furthermore, a number of methods have been developed to demonstrate rainfall uncertainty using rainfall station measurements and to quantify its influence on hydrological simulation results. To demonstrate rainfall uncertainty, the most sophisticated method is a geostatistical approach, the conditional simulation, which is frequently applied to generate the spatial distribution of rainfall by constraining a stochastic dispersion field with the measurements at one, a subset or all of

the rainfall stations. Vischel et al. (2009) used this method to study the differences of simulated runoff using different simulated rainfall spatial pattern, and concluded that for Hortonian runoffs, conditional simulation performs better than simply kriging station rainfall. Moreover, Renard et al. (2011) used the Bayesian total error analysis methodology to decompose the total uncertainty of runoff predictions into the individual contributions of rainfall, runoff, and model structural errors, where the uncertainty in the catchment average rainfall was characterized using the geostatistical

conditional simulation.

The response of hydrological model parameters can be used to quantify the influence of rainfall uncertainty. Chaubey et al. (1999) studied the variation of calibrated model parameters solely due to the spatial variation of rainfall, and a wide range of estimated parameters was found when the rainfall measured at each station was used individually. To alleviate the influence of rainfall spatial uncertainty, Blume et al. (2007) proposed an event-based runoff coefficient in combination with simple



statistical models that could improve the understanding of the rainfall-runoff response of catchments with sparse data. Reichert and Mieleitner (2009) proposed a time-dependent rainfall multiplier at the input side of a hydrological model, which was adjusted according to the good fit of model results.

On the other hand, the influence of rainfall input on hydrological simulation can be investigated by quantifying model

structural errors. The foundation of this method is the assumption that the model structure error represents model limitations and input uncertainty, then the effect of rainfall uncertainty can be explained with the changes in the model bias. For example, Del Giudice et al. (2015) used a Bayesian framework to analyse five configurations of the EPA-SWMM model with different distribution of reservoirs and conduits and different numbers of parameters for sewer flow simulations, and the results showed a progressive decrease of bias with increasing model parameterization, but with an upper bound of model

performance caused by rainfall input.

All the above studies have been conducted to investigate the spatial variation of rainfall, as well as its effect on hydrological simulations. However, different river basins or different rainfall events may lead to different results of spatial rainfall variability, and the effective representative area of rainfall stations could vary remarkably. The results of spatial rainfall variability in a certain case are usually not applicable to other river basins. Therefore, a simple methodology to estimate the

influences of spatial rainfall variability, which can play an important role in understanding the accuracy improvement of hydrological simulation, is necessary.

To this end, first, the bootstrap method, which has the advantages of simplicity (i.e., no need to make any assumption of normality) and high-accuracy (i.e., asymptotically more accurate than the results obtained using sample variance and assumption of normality) (e.g., Chen et al., 2016; García-Soidán et al., 2014; Kim et al., 2004; Legates and McCabe, 1999;

Li et al., 2010b; Uboldi et al., 2014), was introduced in this study to quantitatively analyse the spatial uncertainty of rainfall in river basins at different scales. Second, by employing a distributed hydrological model on a high performance computing (HPC) system, the uncertainty of simulated runoff was also analysed using the bootstrap method. Finally, the response of hydrological simulation to rainfall spatial uncertainty can be formulated by an asymptotic line on rainfall station density.

This paper aims at estimating the influence of rainfall spatial uncertainty in hydrological simulations, focusing on the

spatially distributed rainfall stations. Following this Introduction, this paper has been divided into four further sections: Section 2 introduces the methodology, Section 3 describes the case study catchment and hydrological model, Section 4 presents the results of the uncertainty analysis of rainfall and simulated runoff, as well as related discussions, and Section 5 draws several conclusions.

## 2 Methodology

### 30 2.1 Bootstrap method

The bootstrap method is a resampling method that can refer to any test or metric that relies on random sampling with replacement to quantify uncertainty (Hesterberg et al., 2005). When the bootstrap method is used to estimate a sampling





distribution, the procedure consists of three steps: sampling with replacement, calculating the bootstrap distribution, and using the bootstrap distribution. Resamples from a sample represent what would be obtained if we took a number of samples from the population. Based on the resamples, the bootstrap distribution of a statistic represents the sampling distribution of the statistic. A great advantage of the bootstrap method is its simplicity, which can be a straightforward way to derive

estimates of standard errors and confidence intervals for complex estimators of complex parameters of the distribution. Moreover, it is an appropriate way to control and check the stability of the results, which can be asymptotically more accurate than the standard intervals obtained using sample variance and assumptions of normality (DiCiccio and Efron, 1996). Compared to other resampling methods such as jackknife, the bootstrap method can be successfully used to estimate the non-smooth parameters and the variance of non-linear statistics (Efron, 1979). It is worth noting that parametric methods

are used when we know that the population is approximately normal, or we can approximately use a normal distribution after we invoke the central limit theorem. In contrast, for those we do not know whether the population is normal, or we are not prepared to make any assumption of normality for the population, we use non-parametric methods, for instance, in this study. Indeed, such methods do not have any dependence on the population of interest.

The bootstrap method has been widely applied in hydrological data analysis. For example, Vogel and Shallcross (1996) used

the moving blocks bootstrap to resample the observed time series of stream flow to estimate the storage capacity of a reservoir with different reliability. The bootstrap method can also be applied in time series prediction. For example, Lall and Sharma (1996) proposed a nonparametric nearest neighbour bootstrapping algorithm to find historical nearest neighbours of a current feature vector, a short subset of a time series, and to make predictions by resampling the successor of the current feature from the neighbouring feature vectors' successors. Moreover, Fortin et al. (1997) introduced a non-parametric

Bayesian simulation method, which is asymptotically equivalent to the bootstrap method, but it does not assume that the finite reference sample is equivalent to the whole population, which is assumed by the bootstrap method but not always guaranteed.

Different distributions of rainfall stations may lead to different areal rainfall estimation and runoff simulation. To keep a smaller range of result variation and to reduce the resample times, the bootstrap method in this paper is limited to the

resampling of the presence of each rainfall station at its own location, without location permutation or random placement of the rainfall stations. Therefore, the uncertainty of spatial rainfall and its hydrological response to be quantified is primarily focused on the density of rainfall stations. A bootstrap method with location permutation or random placement can fully explore the spatial uncertainty of rainfall, but the influence of rainfall station density is much more simple and practical to investigate.

In this study, the Thiessen polygon method (Thiessen, 1911) was applied to calculate the average basin rainfall with resampled stations in a river basin. To make up the resampled station sets, two layers of loops were proposed as described in Figure 1. The outer loop is on the number of selected stations, while each inner loop resamples a possible combination of $n$ stations from $K$ groups. The inner loop traverses all the combinations of station groups, including the number of selected groups and the number of selected stations in each group, to form the $n$-station set. To reduce the repetition of simulations,



for a certain combination, a certain number of stations in a certain group are randomly selected only once. By using the bootstrap method, $C_N^n$ rainfall station combinations will be obtained from the real rainfall station set with a resample number of $n$ (Figure 1) and the total number of combinations is $2^N$. To improve the calculating efficiency for the bootstrap method, a python script was composed and executed (Appendix 1).

**2.2 Framework methodology**

Firstly, rainfall events which can reveal different patterns of rainfall-runoff processes at different magnitudes in designated river basins will be selected as the study cases (e.g., Table 1). For each rainfall-runoff event, the rainfall stations in the river basin will be classified into groups. Rainfall processes with the similar time periods, duration, and the time-to-peak can be grouped together (Sivakumar and Woldemeskel, 2015). Based on the total depths and processes of rainfall measurements,
two steps are conducted in this paper using clustering and correlation methods. The first one is based on the total rainfall depth using the clustering method proposed by Park et al. (2009). The other step is the correlation statistics for the rainfall processes of every pair of stations.

Subsequently, reduced numbers of stations is obtained by resampling. Taking the classification of the stations into consideration, the bootstrap method is used to traverse most of the combinations of rainfall stations, and to quantify the
uncertainty of average basin rainfall. Then the influence of the density of rainfall stations on the uncertainty of the spatial rainfall can be analysed using different reduced numbers of stations in the studied river basins.

Finally, after calibrating a hydrological model with all the rainfall stations and measured runoff depth and peak discharge at the river basin outlet, the impact of rainfall uncertainty on hydrological simulation can be assessed using the repeatedly simulated runoff with different rainfall inputs from bootstrapped rainfall, i.e., using different combinations of different
reduced numbers of rainfall stations in the river basin.

**2.3 Digital Yellow River Integrated Model**

The Digital Yellow River Integrated Model (noted as DYRIM hereafter) is a physically-based, distributed-parameter, and continuously-simulated hydrological model developed by Tsinghua University for hydrological and sediment simulations in river basins based on high-resolution digital drainage networks (Wang et al., 2007, 2015). Former studies focused on the
topic of hydrological simulation and soil erosion on the Loess Plateau using the DYRIM, as well as the parallelization of the DYRIM based on dynamic basin decomposition (Li et al., 2011; Wang et al., 2011; Zhang et al., 2016). Moreover, the DYRIM has been widely employed to the simulations of streamflow, floods and sediment, which demonstrated its applicability with satisfactory results in the temporal scales ranging from daily to monthly in the Yellow River basin (Shi et al., 2015, 2016).
The basic unit of the model is river reach and its corresponding hillslopes. All the basic units of a river basin constitute a drainage network (Li et al., 2010a), which is generally extracted from digital elevation model (DEM) data (Bai et al., 2015), e.g., the 30-m-resolution Advanced Spaceborne Thermal Emission and Reflection Radiometer (ASTER) Global DEM



dataset (ASTER GDEM Validation Team, 2009, 2011) used in this paper. Runoff yield is simulated on each hillslope using a two-layer soil model to reflect both the infiltration-excess and storage-excess mechanisms, in fine time steps (e.g., 6 minutes). Infiltration-excess runoff on the hillslope surface, along with related processes, such as vegetation interception, evapotranspiration, ground water discharge and water redistribution between the two soil layers, constitute the fundamental

hydrological processes simulated by this model. Then the flow routing is simulated over this drainage network using a diffusive wave method. The parameters of the DYRIM can be divided into physical parameters that are used to describe the properties of the land use and soil types and determined from field measurements, and adjustable parameters that must be calibrated before model application (Li et al., 2009). And the influence of topography on rainfall is negligible in DYRIM. Several representative adjustable parameters are listed in Table 2.

The topsoil initial moisture ($\theta_{u,0}$) is the initial state of the topsoil layer and $\theta_{d,0}$ represents that of the subsoil layer. The vertical conductivity of the topsoil layer, which is a non-linear function of $K_{zus}$ (the vertical saturated conductivity of the topsoil layer) and $\theta_u(t)$ (the topsoil moisture, $t$ is time). $K_{zus}$ controls the surface infiltration rate and primarily influences the infiltration-excess runoff, as well as $K_{u-ds}$ that controls the rate of water redistribution between the two soil layers. $K_{hu}$ and $K_{hd}$ control the lateral subsurface flow rate of the two soil layers, respectively. The two vertical saturated conductivities ($K_{zus}$

and $K_{u-ds}$) are the key parameters to be calibrated before model application (Wang et al., 2007; Li et al., 2009).

When the DYRIM is applied to a large river basin, high performance computing (HPC) technique is generally used. To implement the bootstrap method in this study, an HPC system with a large number of processor cores is used to guarantee the simulation efficiency. Therefore, the high-speed hydrological simulation of the DYRIM provides the support for repeated simulations in this study.

**2.4 Calibration of the DYRIM**

The calibration of model parameters is parallelized with a double-layer structure on an HPC system (Zhang et al., 2016). A dynamic sub-basin decomposition method (Li et al., 2011) was developed to parallelize the hydrological simulation of the DYRIM, which contributes to the lower-layer parallelism. The MPI standard is adopted to realize the lower-layer parallelism, mainly because it is the dominant technique to develop parallel programs on distributed memory systems that most HPCs

belong to. The job scheduling functions of an HPC system are used to manipulate simultaneous model executions with different parameter combinations in the same generation of an optimization algorithm, which contributes to the upper-layer parallelism.

In this study, the genetic algorithm (GA) (Holland, 1975) was adopted to search optimal model parameters because of its stability, natural parallelism and problem-independence. The GA implementation employed in this paper treats the

parameters to be optimized as real numbers with simulated binary crossover and real-parameter mutation. This technique promises the parameters independent of the GA and easy to be optimized. When the GA needs to evaluate the fitness of various model parameter combinations, the job scheduler of the HPC system is called to put a number of DYRIM jobs with different parameter values into the job list of the HPC and to monitor the job list. When all of the jobs are completed, the





model efficiency will be estimated using observed data to propose the fitness evaluation list. Generations of the GA will be run to explore more parameter combinations until the stop criterion is reached (Zhang et al., 2016).

## 3 Application

### 3.1 Study areas and the rainfall cases

In order to compare the uncertainties of rainfall input in different densities of rainfall networks, two river basins are selected as study areas (Figure 2). The first one is the upstream of the Qingjian River basin (109º12' - 109º44' E, 37º01' - 37°19' N), a tributary of the Yellow River. The second one is the Longxi River basin located in Sichuan province (103º30' - 103º36' E, 31º01' - 31°11' N), a tributary of the Min River that flows into the Yangtze River. The digital drainage networks of these two river basins are also shown in Figure 2.

The upstream of the Qingjian River has a drainage area of 930 km$^2$, and the length of the main stream is 55.6 km. With a semi-arid warm temperate continental monsoon climate, the annual rainfall in this region is 486 mm, and short-duration, high-intensity torrential rains that occur in the flood season (from July to September) account for 65% of the annual rainfall. The basin elevations vary from 930 m to 1562 m, with the obvious declining of terrain from northwest to southeast; but less impact of the variation of elevation on the rainfall distribution have been found. The majority of the basin surface material is

composed of highly erodible loess soil, covering 92% of the basin area. Runoff in this river basin, mainly in the form of infiltration-excess, can cause severe soil erosion that has been investigated in former study (Wang et al., 2007; Shi and Wang, 2015). The total urban areas for resident, mining, industry, and transportation cover 1.72% of the whole basin area, which indicates that the impervious areas have less impact on the runoff yield in this basin. The only hydrological station (i.e., Zichang) is located near the basin outlet, and there are 11 rainfall stations located within or adjacent to this river basin, with

an average controlling area of 84.5 km$^2$ (see Figure 2). All of the rainfall and hydrological data used in this paper were provided by the Hydrographic Bureau of the Yellow River Conservancy Commission. In this study, most of the time steps of the original rainfall data records and the runoff data around the peak were 2 hours and 6 minutes, respectively.

Three rainfall-runoff events representing typical rainfall spatial patterns in the catchment after the year 2000 are manually chosen for the upstream of the Qingjian River basin (see Table 1). These three events differ in the magnitude of measured

peak discharges and in runoff coefficients. The 2002 rainfall event has the heaviest rainfall of 145 mm, leading to a flood with peak discharge of 4670 m$^3$/s and a runoff coefficient as high as 0.600. The other two events have moderate rainfall, with similar basin average rainfall depths calculated from station measurements (34 mm and 28 mm). However, the two events have quite different peak discharges of 881 m$^3$/s and 76 m$^3$/s, respectively, as well as the runoff coefficients of 0.137 and 0.055. These two events were selected because their different rainfall-runoff responses are partially caused by the different

spatial pattern of rainfall (Figure 3), though they cannot be easily simulated accurately using the same set of parameters.

There are 7 rainfall stations located in the Longxi River basin (Figure 2), which has a drainage area of only 80 km$^2$; and therefore, the average controlling area is only 11.4 km$^2$ per station, much smaller than that of the upstream of the Qingjian



River basin. One rainfall event occurred on 2012.7.11 in this river basin is chosen for analysing the areal representative of rainfall stations. The heaviest, lightest and average basin rainfall recorded at the 7 stations during this event were 52.5 mm, 36.5 mm and 47.1 mm, respectively.

**3.2 Runoff simulation**

The time steps of historical rainfall records are different among stations, varying from minutes to several hours. If the stations with high temporal resolution are selected by the bootstrap method, the simulation will benefit from the more detailed rainfall process and then perform better. Because this paper aims at the spatial uncertainty of rainfall, to avoid the influence of temporal resolution on simulation results, the measured rainfall data were firstly pre-treated to have a uniform time step. In this study, the time steps of all the rainfall data were adjusted to the same one, i.e., 2 hours. For rainfall records

with smaller time steps, an accumulation processing was conducted. To deal with a small number of rainfall records at locations with spare time steps, a linear downscaling approach was applied using data from neighbouring stations if these were available, or using a disaggregation of sparser rainfall data into unique 2-hour time steps otherwise. It should be noted that this pre-treatment will affect the precision of hydrological model simulation, mainly because of the aggregation of fine time step rainfall data in this infiltration-excess dominated region.

Then, the DYRIM hydrological model was calibrated using the rainfall data from all the stations. There are two schemes of the model calibration. One is the independent calibration on each rainfall-runoff event, and the other is the calibration on all the events sharing a single set of parameter combination. When the model is calibrated on the events independently one by one, more precise results would be obtained to show the potential performance of the model. However, in such a scheme, the optimized parameters will adapt to each described distribution of rainfall by the stations. Therefore, to insulate the effect of

rainfall spatial distribution from model parameters, at least in an average sense, the three events should be considered comprehensively in model calibration.

A set of adjustable parameters (Wang et al., 2009) were calibrated according to the measured runoff process, runoff volume and peak discharge at the basin outlet (Fares et al., 2014). The precision of simulated runoff process is generally expressed by the Nash-Sutcliffe coefficient of efficiency (noted as NSE hereafter) (Nash and Sutcliffe, 1970):

$$\mathrm{NSE} \; = \; 1 - (\textstyle\sum_{i=1}^{n}(O_i - C_i)^2) / \sum_{i=1}^{n}(O_i - \bar{O})^2, \tag{1}$$

where $C$ is the simulated data, $O$ is the measured data, and the subscript $i$ represents the sequential number of the simulated and measured data series. The NSE approaches 1.0 if the simulated values are quite close to the measured values. A positive NSE indicates that the means of the model predictions and measured values are still close, while a non-positive NSE indicates that the simulated values cannot be strong predictors of the measured sequence.

For runoff process, the time step of the simulated results is 6 minutes, consistent with that of the observed runoff during the main process; however, the time step is much larger in base flow periods. The NSEs were calculated according to the observed runoff data points, basically in the 6-minute time step. But when the observed data points were sparse and did not





exactly match the regular steps of the simulated results, the simulated discharge which was nearest to the actual observation time was selected for each observed data point. Finally, all the selected simulated data points constituted the data series with the observed data points to calculate the NSEs. It is worth noting that the purpose of using the high time step (i.e., 6 minutes) in this study was to emphasize the main runoff process, which led to certain side effects (e.g., a smaller NSE value than usual,

despite the fact that base flow is generally stable and easier to fit).

To demonstrate model performance in representing the rainfall-runoff events, the simulation results during calibration are shown in Table 1, and the simulated runoff processes are compared with observations in Figure 4. Optimized values of the key parameters are shown in Table 2. Because soil hydraulic conductivity parameters are the most sensitive to runoff simulation, only the topsoil vertical saturated conductivity $K_{zus}$ was further optimized in independent calibration.

It can be seen from the simulation results, the volume, peak, time to peak, and fluctuation of the runoff processes were reproduced by the simulations. However, when all the events sharing a single set of parameter combination, the results could not match the observations well, especially in peak discharge. To balance the peak discharges among the three events, there must be one discharge overestimated and another underestimated. The main cause of the simulation errors would be the different spatial pattern of the rainfall events and the large time step of rainfall records. For the spatial pattern, using the

simple Thiessen polygon method in this paper, the distribution can be represented better when real rainfall is more evenly distributed in the river basin; otherwise large errors are expected. For the time step, because each rainfall data point with a 2-hour time step is uniformly assigned to its corresponding 20 simulation time steps of 6 minutes, the real peak intensity of rainfall was under-expressed, resulting in non-realistic vertical infiltration parameters, as well as the early or delayed time to peak. The influence degrees of spatial pattern and time step were different among the simulations of different events; as a

result, a single set of parameter combination could not support all the events well.

In order to demonstrate the potential performance of the DYRIM, the topsoil vertical saturated conductivity $K_{zus}$ was calibrated on the three events independently one by one. The results are shown in Table 1 and Figure 4. In Figure 4, the dash lines of the independently calibrated runoff processes can basically overlap the points of the observations, especially for the 2002 and 2006 rainfall events. With reference to the NSE of flow discharge, it is observed that the NSE values of these three

rainfall-runoff events all significantly improved, from 0.48, 0.43 and 0.05, to 0.61, 0.81 and 0.80, respectively. Such results proved that the DYRIM can represent these rainfall-runoff events in a sufficient way. It is worth noting that the impact of model calibration on results (i.e., the NSE) is sometimes more significant than the influence of rainfall spatially uncertainty on results. Because the runoff of the 2006 rainfall event is much smaller than the other two events with the measured peak discharge of 76 m³/s, the NSE value changed more significantly when the model was independently calibrated.

The deficits of the simulation results are caused by multiple reasons. Except for the aforementioned low temporal resolution of rainfall and different orders of magnitude of the events, there must be some shortcomings in the model structure to meet all the difficulties in hydrological simulation in the middle Yellow River basin. More complex and sensitive hydrological processes occur in semi-arid region than in temperate and wet climates, and some aspects have not been considered in the DYRIM. For example, surface roughness and vegetation of a hillslope affect the residence time of water on its surface, and





finally influence the total amount of infiltration, which is not formulated in DYRIM mainly because of its steep slope assumption. Nevertheless, rainfall records with higher spatial and temporal resolutions are necessitated to better simulate such short-duration and high-intensity rainfall-runoff events. As a result, the analysis of rainfall uncertainty could be conducive to better understanding the rainfall spatial characteristic and its influence on model simulation.

**4 Results and discussion**

**4.1 Spatial variation of measured rainfall**

The results of classification for the three rainfall events are shown in Table 3 and Figure 3. The stations in the same group are shown in the same colour. There are 8 groups of stations for the 2001 rainfall event, 6 groups for the 2002 and 2006 rainfall events.

The 2002 rainfall event is taken as an example to interpret the classification results. It is observed that the stations are mainly in group 2b. However, Zichang station (in group 2a) and Hecaogou station (in group 2f) have significant differences with the others. The total depth in Zichang station is much larger than those in the other stations, and the rainfall process during 7/4 12:00 to 7/5 0:00 of Hecaogou is different from those of the other stations (see Figure 5). The 2002 rainfall event was a heavy rain covering the whole basin, and the measured data of 11 stations show statistically significant differences.

The classification results of these rainfall stations show that, generally, in a rainfall event, some stations have the statistically uniform rainfall process and total depth, but those of the other stations are significantly different. Moreover, the stations in the largest group are generally close to each other. The stations in group 2b of the 2002 rainfall event are mainly located in the north of the basin, except for Xinzhuangke; the stations in group 3a of the 2006 rainfall event are mainly located in the west of the basin, except for Zichang.

The station that recorded the highest rainfall is regarded as the rainfall centre. Total depth recorded in the rainfall centre is generally 2-5 times greater than the average basin rainfall. With the current density of rainfall stations (noted as $D_S$), the rainfall centre usually involves only 1-2 stations, such as Jingzeyan station in the 2001 rainfall event, Zichang station and Hecaogou station in the 2002 rainfall event, and Zichang station in the 2006 rainfall event. That is, the rainfall centre can only be captured with an approximately 100 km$^2$ resolution at this density of rainfall stations. However, the coverage area of

the rainfall centre may be smaller than that scale. Therefore, the data recorded in the rainfall centre will have an important influence on basin rainfall as well as runoff simulation.

**4.2 Uncertainty of basin rainfall**

Uncertainty must exist when 11 rainfall stations are used to measure the basin rainfall in a 930 km$^2$ area. In this study, the DYRIM is a distributed hydrological model, with those multiple rainfall stations, the uncertainty of rainfall cannot be easily

quantified by observing the variation of rainfall input multipliers for optimal results, following the method used by Reichert and Mieleitner (2009) with a lumped conceptual hydrological model and then a single rainfall multiplier. Even if the





DYRIM model is used with a single areal averaged rainfall input, the time cost of the thousands of repetitive model runs is not acceptable. Therefore, it is difficult to evaluate the original uncertainty range of a certain rainfall event with the actual number of stations, noted as $U_0$, and it will not be performed in this paper.

We focus on the change of uncertainty range corresponding to the number of rainfall stations. The increased uncertainty 5 range, noted as $U(n)$, can be obtained by resampling $n$ rainfall stations from the actual $N$ stations. To make comparisons among different storm events and river basins, a non-dimensional uncertainty range $U'(n)$ is also derived in this paper. However, because of the lacks of true rainfall depth and the original uncertainty range $U_0$, the $U'(n)$ can only be calculated as $U'(n)=U(n)/R_0$, where $R_0$ is the basin average rainfall using all the stations. It should be noted that the non-dimensional uncertainty range $U'(n)$ cannot be inferred as the uncertainty range relative to the original, $U(n)/U_0$, nor the difference from 10 the original, $U(n)-U_0$. The $U'(n)$ is only used to demonstrate the trend of uncertainty range over each resampled number of rainfall stations $n$.

The uncertainty ranges are reflected by box plots. A box plot is a description of the data distribution, which denotes the five percentiles of the performance of quantitative variables, i.e., $P_{2.5}$, $P_{25}$, $P_{50}$, $P_{75}$, and $P_{97.5}$. The $P_{25} - P_{75}$ range constitutes a box of the graphics, and the $P_{2.5} - P_{25}$ and $P_{75} - P_{97.5}$ ranges constitute the two whiskers. The lowest and highest datum are still 15 within a 1.5 interquartile range, and any data out of the whiskers should be plotted as an outlier with a dot or small circle. The $U(n) - n$ and $U'(n) - n$ results of the rainfall and simulated runoff of the three rainfall-runoff events are shown in Figure 6.

For the 2001 and 2006 small rainfall events, the outliers of basin rainfall and simulated runoff depth occur frequently as the $n$ values are greater than 6, and the uncertainty ranges become much larger as the $n$ values are less than 6. For the 2002 heavy 20 rainfall event, the outliers of basin rainfall and simulated runoff depth appear evenly, and the uncertainty ranges are smaller than those of small rainfall events. This phenomenon could be caused by the smaller size of rainfall centre for small rainfalls, i.e., the size of the small rainfall centres is much smaller than the area controlled by each rainfall station.

**4.3 Areal representative of rainfall stations**

Based on the results of the upstream of the Qingjian River basin with resampled rainfall stations, the range of the average 25 basin rainfall shows wide fluctuation. What occurs if the coverage of rainfall stations in a basin is denser? The rainfall event (average basin rainfall 47.1 mm) in the Longxi River basin with a denser rainfall station network and the 2001 rainfall event (average basin rainfall 34.2 mm) in the upstream of the Qingjian River basin were compared (see Figure 7). The horizontal axis is the mean representative area of a single station that calculated from a certain number of the resampled stations.

The results show the fluctuation range of average basin rainfall expands when the representative area of a single station 30 increases. However, even if only one station exists in the Longxi river basin, the average basin rainfall fluctuates within a 25% range, but the ranges of the upstream of the Qingjian river basin are 2-3 times wider. When the representative area of a single station decreases to less than 40 km$^2$, the fluctuation range of average basin rainfall has no evident changes. Thus, the uncertainty range of average basin rainfall has strong correlation with the station density in a river basin. With increasing





station density, the uncertainty of average basin rainfall decreases. However, in a real-world situation, the station density has an affordable threshold at which the uncertainty could not further be decreased efficiently for certain applications and costs.

**4.4 Uncertainty of simulated runoff**

After calibrating the model with a single set of parameter combination, runoffs were repeatedly simulated using different

numbers of resampled rainfall stations with different station combinations. Then the uncertainty of simulated runoffs was analysed in regard of rainfall station density. From Figures 6(c) and 6(d), it can be observed that the simulated runoff depth using resampled partial stations presented a larger range of non-dimensional uncertainty than average basin rainfall, as well as the fluctuation of the mean. That is, the uncertainty of the rainfall input is amplified by the rainfall-runoff process, resulting in greater uncertainty of the simulated runoffs.

Figure 8 illustrates the relationship of the NSE with the number of stations, as well as the ratio of simulated runoff depth over measured. With the reduction of rainfall input information, the mean of the NSE decreases and its range expands. The average level of the simulated runoff volume fluctuates with no significant trend, but the expansion of the range is still notable. Meanwhile, as shown in Table 4, the non-dimensional uncertainty range ($U'$) of simulated runoff depth is 2 to 7 times wider than that of average basin rainfall, for a certain number of stations. That is to say, the hydrological simulation is

sensitive to the spatial uncertainty of rainfall, and slight differences in rainfall input may lead to greater variance of simulated runoff. This amplification can probably be explained by the infiltration-excess rainfall-runoff mechanism (Shi et al., 2016; Wang et al., 2007). The fluctuations of total rainfall and excess rainfall are similar in depth. But the depth of excess rainfall is smaller, and then the fluctuation rate (in percentage) of runoffs (generated by excess rainfall) is much larger. The effect of rainfall centre in each rainfall event is analysed as follows. For the simulation of the 2001 rainfall event, the

NSE (Figure 8(a)) reduced to 0.18 (using 10 stations) from 0.48 (using 11 stations), in the absence of Jingzeyan station with the highest rainfall depth of 75.2 mm (Figure 3(a)). For the 2002 event, the NSE (Figure 8(c)) reduced to 0.26 (using 10 stations) from 0.43 (using 11 stations), in the absence of Zichang station with the highest rainfall depth of 283.2 mm (Figure 3(b)). For the 2006 rainfall event, because Yujiawan station (Figure 3(c)), which recorded the highest rainfall depth (46.4 mm), only controls a very small area (0.13%), the rainfall centre is mainly represented by Zichang station, with the second

highest rainfall depth of 42.8 mm. When Zichang station was absent, the NSE (Figure 8(e)) reduced to -0.48 (using 10 stations) from 0.05 (using 11 stations). As a result, it can be concluded that, with certain rainfall input combination, if the rainfall centre is not captured and applied to the hydrological model, a large portion of the actual rainfall will be ignored in the rainfall-runoff simulation, resulting in significant deviation in the simulated runoff. Even though some methods can produce high-intensity rainfall cells in a river basin, e.g., the geostatistical conditional approach used by Vischel et al. (2009)

and Renard et al. (2011), the intensity and area of the artificial rain centre are difficult to simulate. Therefore, to obtain more confident simulation results, higher rainfall station density ($D_S$) must be provided to capture more accurate range of rainfall centres.





Furthermore, in the 2001 rainfall event, the NSE of the simulation result is 0.48 using all the 11 stations. However, the NSE increases to 0.51 in the absence of Yangkelangwan station (rainfall depth = 31.8 mm), and reduces to 0.07 in the absence of Sanshilipu station (rainfall depth = 36 mm). It can be concluded that even the rainfall records of different stations are similar to each other (in the same group), their locations have a strong influence on hydrological simulation.

**4.5 Effect of rainfall station density on hydrological simulation**

For the NSE of simulated runoff with different number of rainfall stations ($n$) in Figure 8 (a), (c), (e), when $n$ decreases, the NSE decreases because of the lack of rainfall information. To better quantify this relation, the mean NSE value for each $n$ is correlated with the corresponding rainfall station density $D_S$, i.e., the number of stations in each 1000 km$^2$ area, as shown in Figure 9.

An obvious relationship between the mean NSE of simulated runoff and the rainfall station density can be found in Figure 9, and this relationship can be fitted using an exponential curve (see the solid line). Furthermore, if the fit line is extrapolated to larger $D_S$ values (the dot dash line), the NSE will further increase, implying that the runoff simulation results will be improved. However, even if we can install as many rainfall stations as possible over the river basin, there is a limit of the NSE because of temporal uncertainty of rainfall, as well as the inherent uncertainty of the hydrological model and

parameters. According to this trend, the exponential fitting curve was proposed as:

$$\text{NSE} = a * \exp(-b * D_s) + \text{NSE}_{max}, \tag{2}$$

where $a$ and $b$ are the fitting parameters, $\text{NSE}_{max}$ is the NSE value achieved when the $D_S$ approaching infinity. However, as the NSE value should be no more than 1, an upper limit of $\text{NSE}_{max}$ (i.e., 1) is set when conducting the fitting process (the longer dash line in Figure 9).

In the first two cases, the NSE values were satisfactory and could reach 0.8 if the $D_S$ values were greater than 21 and 26 stations per 1000 km$^2$, respectively. One general inference of this result is that about 40 km$^2$ controlling area is appropriate for each rainfall station for hourly hydrological simulation in this region. This inference is in accordance with Figure 7 that the average basin rainfall estimation fluctuates within a 15% range when the representative area of each station is less than 40 km$^2$. Several studies (e.g., Villarini et al., 2008) also suggested higher spatial resolutions for rainfall measurement, but it

is difficult to build and maintain a denser rainfall station network for the practical and financial reasons in many river basins. In the third case, the NSE value was low, even if only higher $D_S$ values (6 to 11) were used for curve fitting. The spatially improvement of rainfall input could not enhance the simulation performance of this case to NSE greater than 0.2. After further examining the rainfall-runoff events, it was found that the runoff coefficient of the 2006 rainfall event (0.055) is remarkably lower than the former two events (0.137 and 0.600). As the same set of parameters of the DYRIM hydrological

model is calibrated for all the three events, the limitation of hydrological simulation of the 2006 rainfall event was possibly caused by the hydrological model and parameters, rather than the spatial uncertainty of rainfall records. The variation of the





runoff coefficient is also a representation of the difficulty for model simulation in this infiltration-excess region, as mentioned in Sub-section 4.4.

The proposed fitting of Equation (2) for the relationship between the NSE of flow discharge and rainfall station density would be capable to reveal the possible improvement of simulation when the number of stations increases to reduce the
spatial uncertainty of rainfall. Therefore, this bootstrap method is useful in managing rainfall station density, which can give the advice for the proper $D_S$ in a certain river basin. Nevertheless, the deficiencies in model structure and model parameters also substantially contribute to the uncertainties of results in some cases, as shown in Figure 9(c). The proposed bootstrap method could be promising to further quantify the fraction of rainfall input-caused uncertainty from the whole simulation error, after integrating some existing approaches (e.g., Del Giudice et al., 2015).

**5 Conclusions**

This paper proposed a bootstrap method to quantify the spatial uncertainty of rainfall by resampling current rainfall stations. The bootstrap method was further adopted to evaluate the influence of rainfall spatial uncertainty on hydrological simulation, using the DYRIM hydrological model in the upstream of the Qingjian River basin and the Longxi River basin.

For a certain rainfall event, the stations in the studied river basin represented statistically significant correlations, in terms of
total rainfall depth and process, and the stations clustered in the same group were generally in close geographic proximity. The spatial uncertainty of rainfall was introduced as the fluctuations of average basin rainfall using resampled partial stations, and it was found smaller when a larger rainfall event covering the whole basin occurs. The uncertainty of the average basin rainfall between different river basins was found with essential homogeneity, by comparing the studied basin with the Longxi River basin.

The uncertainty of simulated runoff caused by the spatial uncertainty of rainfall was quantified by using the DYRIM hydrological model with the bootstrapped rainfall stations. The hydrological simulation was found sensitive to the spatial uncertainty of rainfall. The uncertainty of rainfall input was amplified remarkably through hydrological simulation, resulting in greater uncertainty of the simulated runoffs. Generally, with the reduction of rainfall input information, the effectiveness of hydrological simulation decreases, and the fluctuation range of simulated runoff increases. Moreover, the data records at
the rainfall centre reflect a higher weight for the simulation.

There should be an optimal layout of rainfall stations (including number and location) in a designated river basin. In this study, the relationship between the NSE of simulated runoff and the rainfall station density in the upstream of the Qingjian River basin was established, and this relationship can be extrapolated to predict the runoff simulation improvement when station density increases, which can be regarded as a suggestion to plan the number of rainfall stations to strengthen rainfall
observations. A precise description of the spatial distribution of rainfall is possible with a very dense distribution of rainfall stations, or by using other techniques. For example, the NSE value could be higher if there were more stations in the upstream of the Qingjian River basin. However, it should be noted that the relationship is proposed using a simple and



straight-forward method, its application is the relation's extrapolation and lacks rigorous theoretical derivation. Further works should be done to rationalise the method and to verify it in a densely measured river basin. Moreover, with reference to the location, additional rainfall stations should be installed as close to the rainfall centre of a river basin as possible for better flow simulation, which can be obtained from the analysis of local historical high-resolution radar-based rainfall data.

5    It should be also noted, the method to estimate average basin rainfall has an influence on the uncertainty analysis. This study employed the Thiessen polygon method, but other methods, such as the Kriging method and the inverse distance weighted (IDW) method, may lead to different results, which need to be further studied, and the results are also easily obtained using the bootstrap method.

**Acknowledgements**

10    This research was supported by the National Natural Science Foundation of China (Grant No. 51459003, 51579131) and the State Grid of China (Grant No. SGQHJY00GHJS1500057). Any use of trade, firm, or product names is for descriptive purposes only and does not imply endorsement by the authors and funders.





**Appendix 1**

The pseudo-codes of the python script functions to calculate the basin average rainfall depth according to the bootstrap method are given as follow. The functions are easy-to-use and high compatible with the ArcGIS 10.0 environment, independent of hydrological processes.

```
import arcpy //Import ArcGIS library

For (n=1, n++, n<=N) //Loop on n, the number of rainfall stations from all the N rainfall stations

    For a certain combination "x"//Loop on each combination of stations, controlled by different file folders.

        arcpy.env.workspace="File_path_combi_x"//Locate the workspace for a certain combination "x".

        arcpy.MakeXYEventLayer_management("StationCoordinate.txt","x","y","Filename_x_y","#","#")
            //Read a prepared file "StationCoordinate.txt" with bootstrapped stations with coordinates "x"
            and "y". All the "File_path_combi_x" file directories and "StationCoordinate.txt" files are
            created by a recursive method.

        arcpy.FeatureToPoint_management("Filename_x_y","management.shp","CENTROID")
            //Create a point shape file from the station file. Output: "management.shp".

        arcpy.env.extent=arcpy.Extent()//Set the range of the area.

        arcpy.CreateThiessenPolygons_analysis("management.shp","analysis.shp","ALL")
            //Thiessen polygons creation. Input: "management.shp", Output:"analysis.shp".

        arcpy.Clip_analysis("analysis.shp","BoundBuff.shp","clip.shp","#")
            //Thiessen polygons clipping with the given basin bound file. Input: "analysis.shp", Output:
            "clip.shp", Bound file: "BoundBuff.shp".

        arcpy.CalculateAreas_stats("clip.shp","stats.shp")
            //Area calculation of each polygon. Output: "stats.shp"

        arcpy.ExportXYv_stats("stats.shp","NO;F_AREA","SPACE","area.txt")

            //Result output of each station's control area, and the areal rainfall could be easily obtained by
            a weighted method using the control area and rainfall records of each station.

    //End of combinations loop
```



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




**Figures**

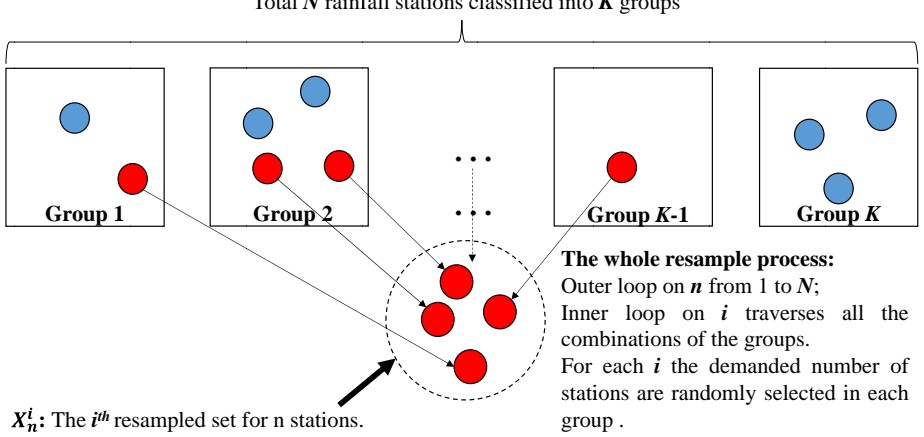

$X_n^i$: The $i^{th}$ resampled set for n stations.

**Figure 1: Framework of the resample processes of the bootstrap method.**

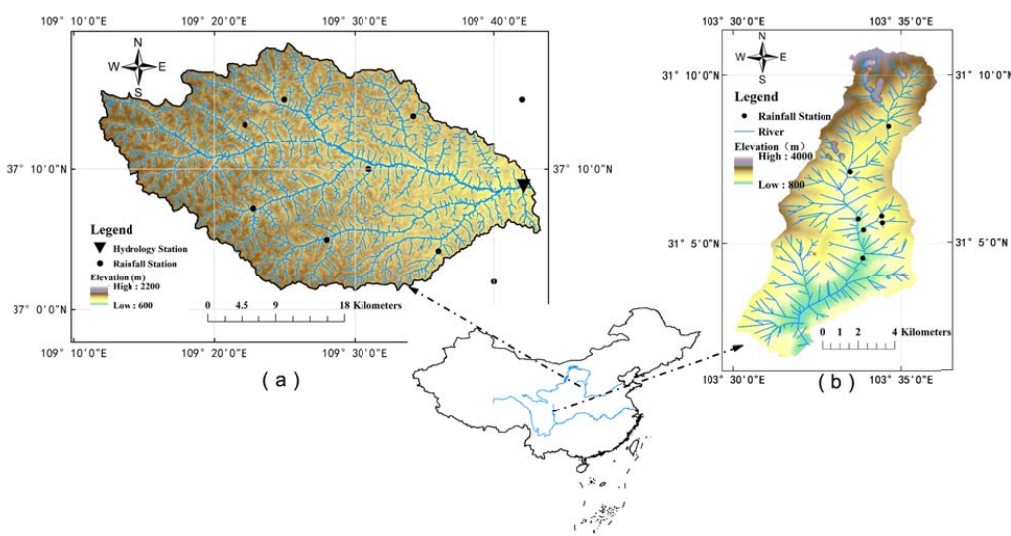

**Figure 2: The distribution of hydrological and rainfall stations in (a) the upstream of the Qingjian River basin and (b) the Longxi River basin.**



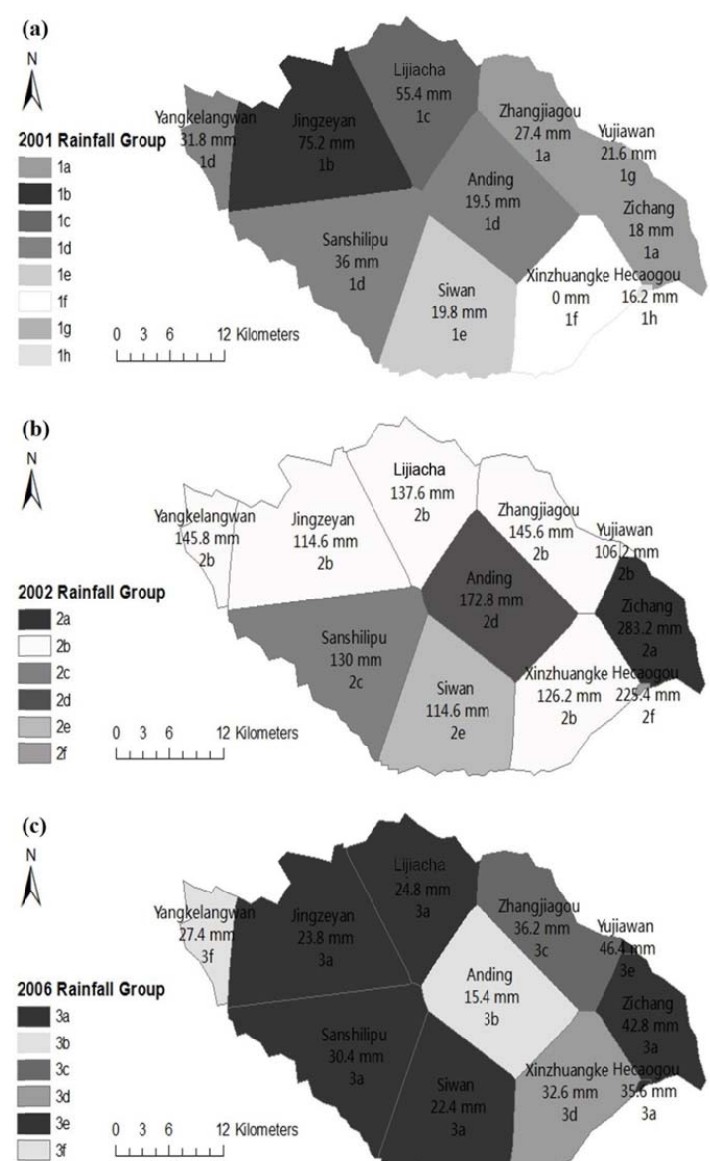

**Figure 3: Classification results of rainfall stations of three typical events: (a) 2001, (b) 2002, (c) 2006. Classification using clustering based on the total depths and correlation methods based on the processes of rainfall measurements are conducted.**





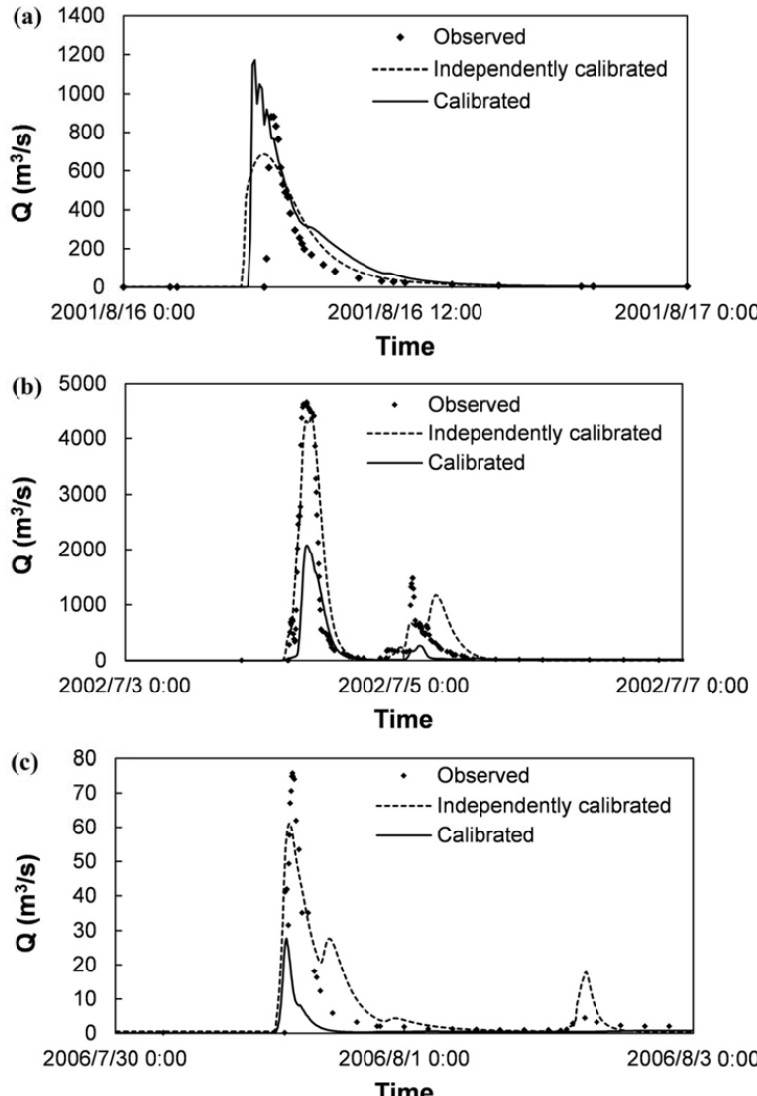

**Figure 4: Calibrated runoff processes compared with observations of the three events: (a) 2001, (b) 2002, (c) 2006.**





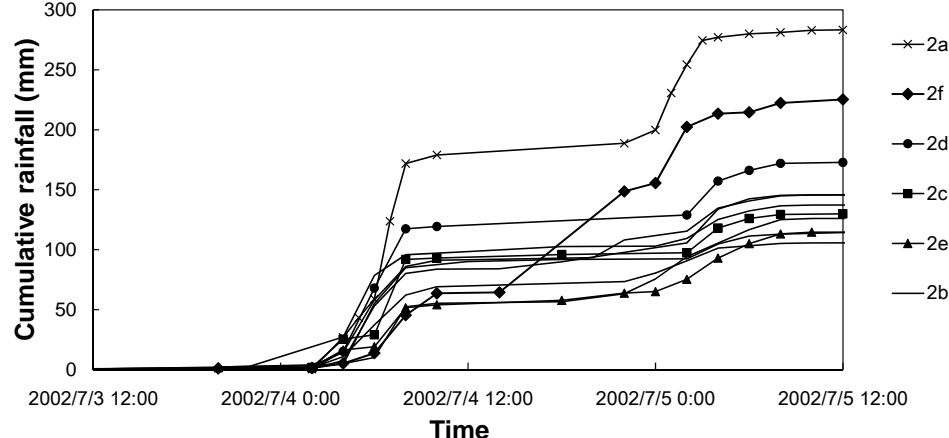

Figure 5: Cumulative rainfall of each station in the 2002 rainfall event.





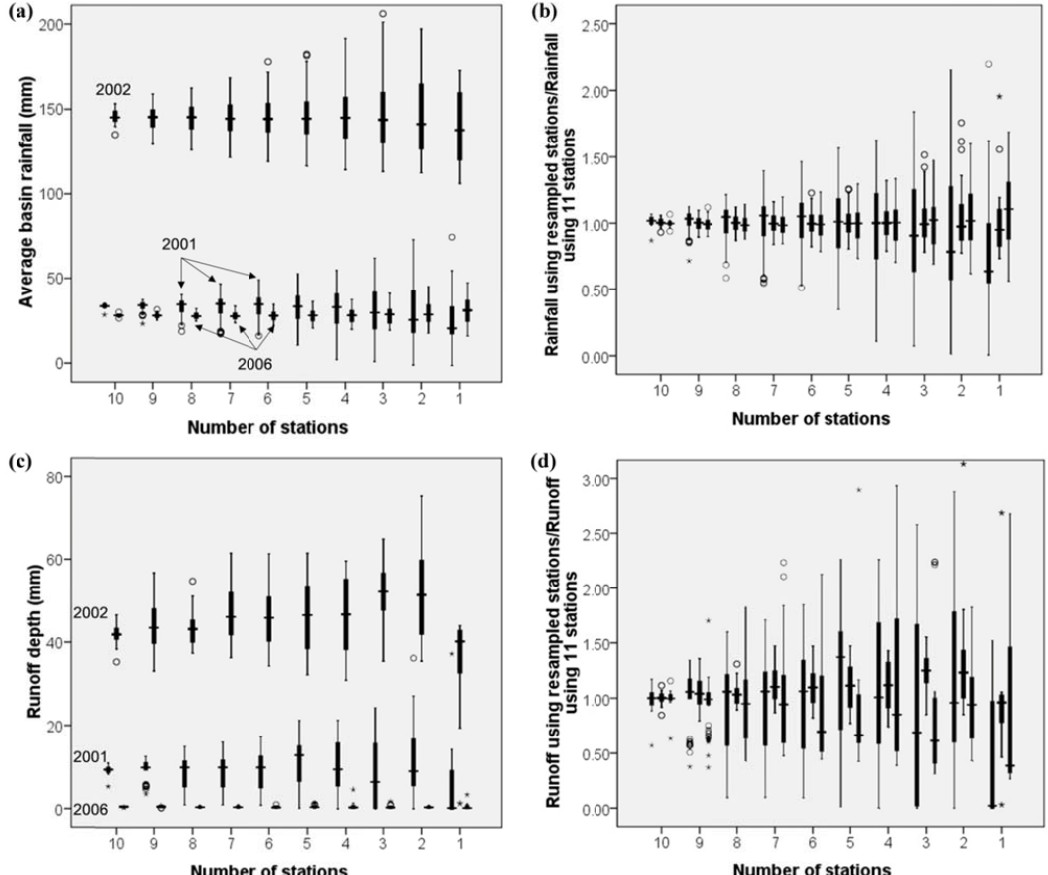

**Figure 6: Uncertainty of the three rainfall-runoff events on different number of stations: (a) Average basin rainfall, (b) Non-dimensional uncertainty range of average basin rainfall, (c) Simulated runoff depth, and (d) Non-dimensional uncertainty range of simulated runoff depth. Every three boxes represent the results of 2001, 2002, 2006 events from left to right.**





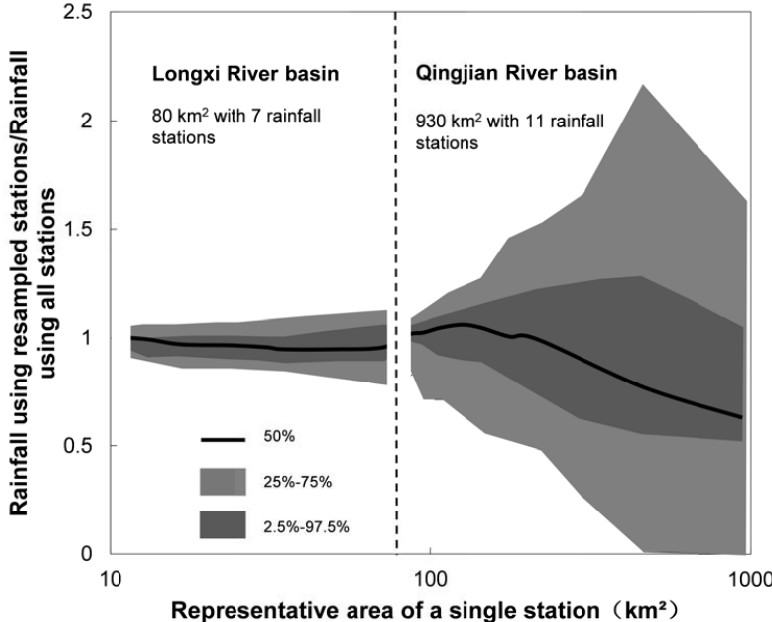

**Figure 7: The non-dimensional uncertainty range of average basin rainfall over the representative area of a single station.**



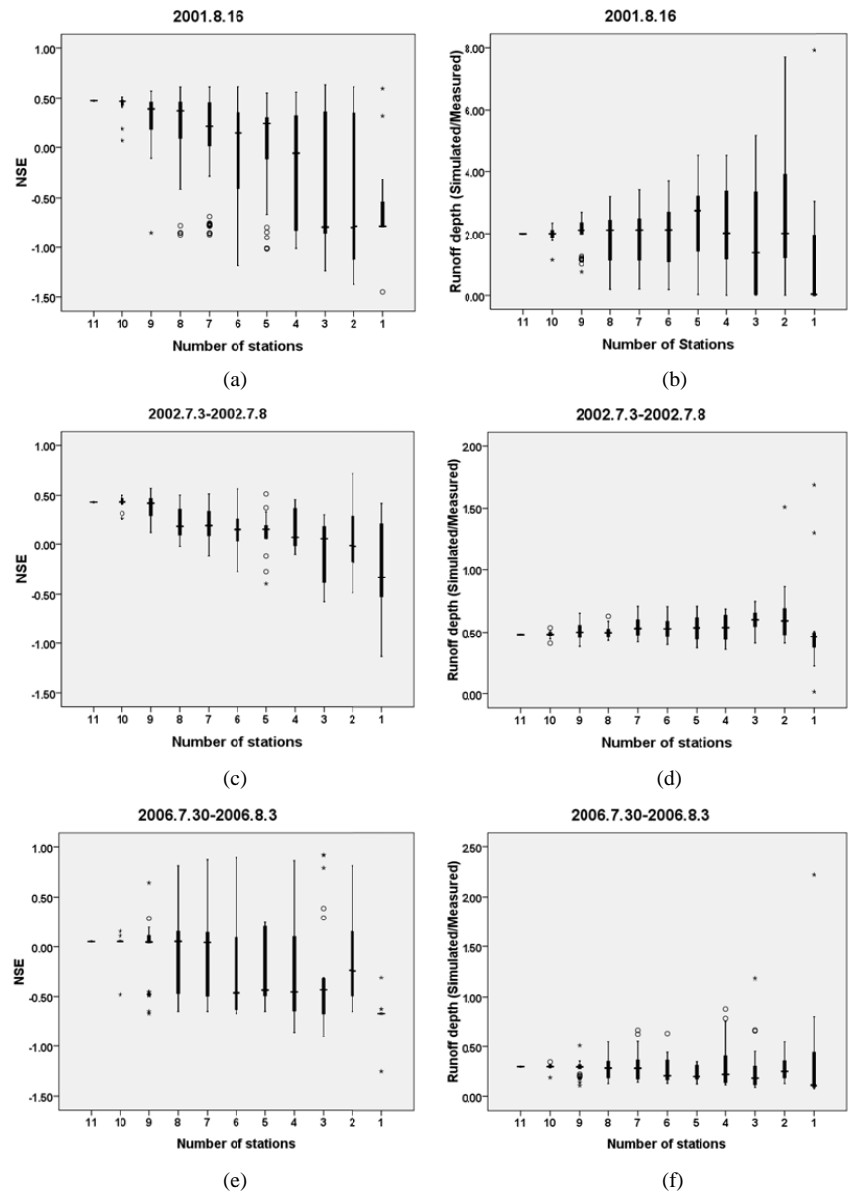

Figure 8: Performance of simulated runoffs of the three cases using different numbers of resampled stations: (a), (c), (e) NSE; (b), (d), (f) the ratio of simulated runoff depth over measured.

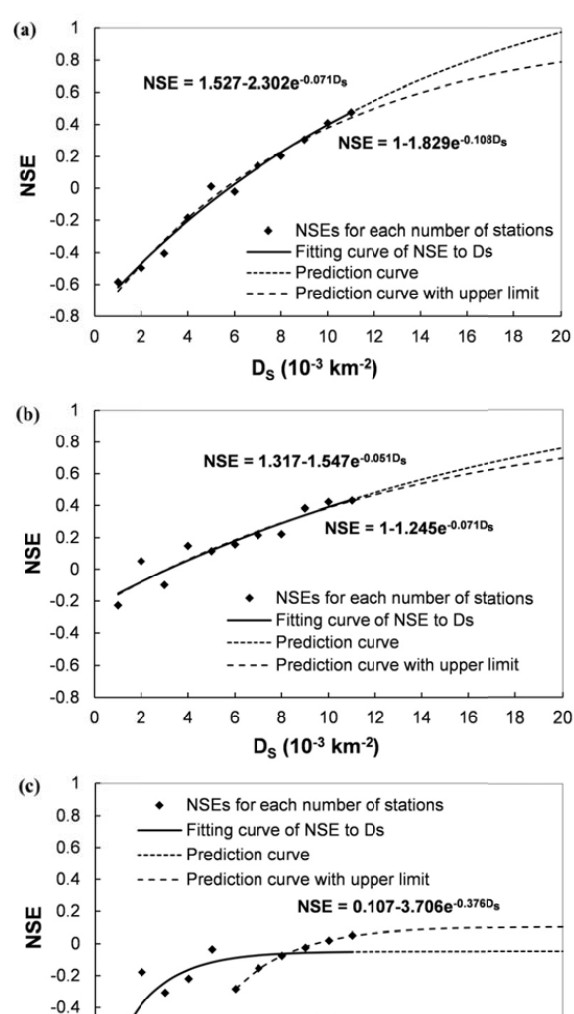

**Figure 9:** Estimation of the effect of rainfall station density on hydrological simulation events: (a) 2001, (b) 2002, (c) 2006. The solid lines are the fitting curve between the NSE of simulated runoff and the rainfall station density and the dot dash lines are the prediction if the DS increase. And the dash lines are the predicted NSE values with the upper limit of 1.





**Table 1: Characteristic parameters of the three rainfall events for the upstream of the Qingjian River basin.**

| Event number | 1 | 2 | 3 |
|---|---|---|---|
| Duration | 2001.8.16 | 2002.7.3-7.7 | 2006.7.30-8.3 |
| Basin average rainfall (mm) | 34.2 | 144.9 | 27.5 |
| Measured runoff depth (mm) | 4.7 | 86.9 | 1.5 |
| Runoff coefficient | 0.137 | 0.600 | 0.055 |
| Simulated runoff depth (mm) | 7.0 | 40.1 | 0.5 |
| Independently simulated runoff depth (mm) | 7.9 | 111.5 | 2.1 |
| Measured peak discharge (m³/s) | 881 | 4670 | 76 |
| Simulated peak discharge (m³/s) | 1172 | 2079 | 28 |
| Independently simulated peak discharge (m³/s) | 686 | 4407 | 61 |
| Measured peaking time | 8/16 6:18 | 7/4 7:15 | 7/31 5:24 |
| Simulated peaking time | 8/16 5:36 | 7/4 7:18 | 7/31 4:24 |
| Independently simulated peaking time | 8/16 6:00 | 7/4 8:06 | 7/31 4:54 |
| NSE of flow discharge | 0.48 | 0.43 | 0.05 |
| Independently calibrated NSE of flow discharge | 0.61 | 0.81 | 0.80 |




**Table 2: Key parameters of the model after calibration.**

| Parameters | Comprehensive calibration | Independent calibration of 2001 | Independent calibration of 2002 | Independent calibration of 2006 |
|---|---|---|---|---|
| Topsoil vertical saturated conductivity $K_{zus}$ (mm/hr) | 3.7 | 6.2 | 0.8 | 3.2 |
| Subsoil vertical saturated conductivity $K_{u\text{-}ds}$ (mm/hr) | | 5.2 | | |
| Topsoil lateral saturated conductivity $K_{hu}$ (mm/hr) | | 5.9 | | |
| Subsoil lateral saturated conductivity $K_{hd}$ (mm/hr) | | 3.6 | | |
| Topsoil initial moisture $\theta_{u,0}$ (m$^3$/m$^3$) | | 0.15 | | |
| Subsoil initial moisture $\theta_{d,0}$ (m$^3$/m$^3$) | | 0.23 | | |



**Table 3: Classification of rainfall stations in the three rainfall events.**

| NO | Station name | Percentage of controlling area(%) | Rainfall records (mm) and group ID of each station | | | | | |
|----|--------------|-----------------------------------|------------|------|-----------------|------|-----------------|------|
| | | | 2001.8.16 | | 2002.7.3-2002.7.8 | | 2006.7.30-2006.8.3 | |
| 1 | Zichang | 7.48 | 18.0 | 1a | 283.2 | 2a | 42.8 | 3a |
| 2 | Jingzeyan | 16.37 | 75.2 | 1b | 114.6 | 2b | 23.8 | 3a |
| 3 | Lijiacha | 11.17 | 55.4 | 1c | 137.6 | 2b | 24.8 | 3a |
| 4 | Sanshilipu | 15.19 | 36.0 | 1d | 130.0 | 2c | 30.4 | 3a |
| 5 | Anding | 12.31 | 19.5 | 1d | 172.8 | 2d | 15.4 | 3b |
| 6 | Zhangjiagou | 10.02 | 27.4 | 1a | 145.6 | 2b | 36.2 | 3c |
| 7 | Siwan | 12.79 | 19.8 | 1e | 114.6 | 2e | 22.4 | 3a |
| 8 | Xinzhuangke | 10.88 | 0.0 | 1f | 126.2 | 2b | 32.6 | 3d |
| 9 | Yujiawan | 0.13 | 21.6 | 1g | 106.2 | 2b | 46.4 | 3e |
| 10 | Hecaogou | 0.09 | 16.2 | 1h | 225.4 | 2f | 35.6 | 3a |
| 11 | Yangkelangwan | 3.57 | 31.8 | 1d | 145.8 | 2b | 27.4 | 3f |
| | Total | 100 | 8 groups | | 6 groups | | 6 groups | |





**Table 4: The non-dimensional uncertainty ranges (U') of average basin rainfall and simulated runoff depth.**

| Event year | U' | Number of stations | | | | | | | | | |
|---|---|---|---|---|---|---|---|---|---|---|---|
| | | 10 | 9 | 8 | 7 | 6 | 5 | 4 | 3 | 2 | 1 |
| 2001 | Rainfall | 0.20 | 0.41 | 0.63 | 0.86 | 0.96 | 1.22 | 1.52 | 1.77 | 2.15 | 2.20 |
| | Runoff | 0.60 | 0.97 | 1.50 | 1.61 | 1.75 | 2.24 | 2.26 | 2.58 | 3.85 | 3.96 |
| 2002 | Rainfall | 0.13 | 0.20 | 0.25 | 0.32 | 0.41 | 0.45 | 0.53 | 0.73 | 0.98 | 1.22 |
| | Runoff | 0.60 | 0.97 | 1.50 | 1.61 | 1.75 | 2.24 | 2.26 | 2.58 | 3.85 | 3.96 |
| 2006 | Rainfall | 0.13 | 0.22 | 0.26 | 0.35 | 0.45 | 0.56 | 0.63 | 0.78 | 0.98 | 1.13 |
| | Runoff | 0.53 | 1.35 | 1.41 | 1.78 | 1.70 | 2.51 | 2.58 | 3.67 | 7.61 | 7.22 |