# Peer review of "A bootstrap method to estimate the influence of rainfall spatial uncertainty in hydrological simulations"

_Hydrology and Earth System Sciences, 2017_

## Referee Comment (RC1) · Anonymous Referee #1 · 3 Jul 2017

GENERAL COMMENTS The authors set out to establish a simple method for estimating the uncertainty of areal rainfall estimates on hydrologic simulations. While such approaches have been broadly considered (and are well cited in the introduction), the authors suggest that existing approaches lack an ability to be extrapolated to other places. Presumably the authors believe their approach addresses this, though that is not explicitly clear to the reviewer. Even if this were the case, unfortunately the modeling study suffers from fatal flaws that prohibit further interpretation of the results. Namely, the authors rely on running a 6-min simulation model with 2-hour forcings (precipitation). This causes multiple problems with model behavior (see specific comments). Additionally, the model clearly suffers from misspecification as a result of over-

fitting – demonstrating that the curve can be matched but at the expense of process fidelity.

SPECIFIC COMMENTS P3.L14-16: Is the argument for this study that a more straightforward methodology is needed? Or simply that this type of study needs to be performed at more sites than it has been to date?

P3.L24-25: In L12-14 on this page, you argue that results are not reliably extrapolated to other locations, so can your study really accomplish this, as stated?

P4.L30-31: Why use Thiessen? Why not use something more sophisticated?

P6.L1-3: Why are you using 6-min time steps here, when DYRIM performance above (P5.L26-29) is noted as satisfactory for daily to monthly time steps?

P6.L6-8: How many parameters are in DYRIM, how many need to be calibrated?

P6.L28-29: Why would you use this rather than something like SCE-UA or DDS, which are recognized to be the "best" global optimization algorithms for hydrological applications?

P9.L3-4: But, if you don't have 6-min forcing data, what does the 6-min runoff simulation get you? I am also concerned by the number of ad hoc adjustments being made in the model implementation (disaggregating rainfall data to smaller time steps, selecting the nearest observed runoff time for NSE evaluation, etc).

P9.L13-14: Exactly, you can't really simulated 6-min runoff with 2-hour rainfall. My question, then, is what value do these results have? The study would be stronger if the data fit the desired framework. . .

P9.L14-16: So why use Thiessen method?!

P9.L16-19: This is a clear flaw in the study design. Any interpretation of these results will naturally be influenced by this.

[Figure]

P9.L21-22: Why should the vertical saturated conductivity of the topsoil change so much? Shouldn't this value theoretically be constant? Aren't these changes representative of compensation of other errors in the modeling?

P9.L25-26: But it doesn't prove that it achieves this performance for an appropriate reason. This is just curve fitting, isn't it?

P10.L6: How would you utilize these results to improve rainfall gauging density or placement? Can you predict these locations without already having measured rainfall? If not, then what is the benefit?

TECHNICAL CORRECTIONS P2.1-2: "more" reliable. . .

P3.3: Change "good fit" to "goodness of fit"

---

## Referee Comment (RC2) · D. Klotz (Referee) · 11 Sep 2017

**General Comments**

The present study proposes a bootstrap based quantification of the spatial rainfall uncertainty for rainfall-runoff modelling. The main body of the work uses aforementioned bootstrap method is used to select between different numbers of stations and different groups of stations. The influence of the number and grouping is then quantified via the Nash Sutcliffe Efficiency and exponential fit is used to derive a heuristic for the estimation of the optimal densities of rainfall measurement stations. The whole analysis is based on three manually selected rainfall events from 2001, 2002 and 2006,

with data from of the Quingjian basin. Additionally, rainfall data from a second basin (Longxi) is used to provide a basic comparison about the scale and uniqueness of the conclusions.

The topic as such is certainly relevant for the study of the rainfall-runoff relationship as such and would fit a HESS. I do also conceive that the endeavor of the authors poses major challenges. Rainfall process can be extremely heterogeneous and as the authors state a quantification of the uncertainties depends multiple interplaying phenomena (such as the position of the meteorological stations, the unique futures of the studied events or the elevation profiles of a catchment). So, the authors did probably a good job, if we consider the possible difficulties that are inherent in such a study. I am nevertheless left with many concerns. The most important wants are listed in the following points, minor comments and questions are given later.

**1.** First and foremost, I believe that considerable attention should be given to improve the introduction in special and the readability of the manuscript in general. The whole manuscript needs a clearer structure, the aims must be defined in a clearer fashion (maybe less focus on different aspects and more focus on the main topics) and the manuscript as a whole should meander less between topics. Arguments about the computational expense, for example, can be found throughout the script, but no chapter is devoted to it.

**2.** In reference to my last point, I have to say that I truly believe that is worth to discuss the computational expense of the method as such. Currently hints about the computational efficacy are spread throughout the manuscript, but I miss a dedicated chapter and quantitative arguments. The method seems quite computationally expensive as the hydrological model needs to be executed approximately N*K times. Now what I would like to know is what it does mean quantitatively. How long does a run of the used r-r model take, and how long does an execution of the bootstrap need. Especially the last point provides a plethora of possible topics/experiment: Influence of shorter and longer time series, usage of the bootstrap without and with parallelization, usage of the

model with and without parallelization, on a cluster or a normal pc, etc.

**3.** The actual time-resolution of the modelling needs some further discussion and explanations. I think the main problem here is that the descriptions are not specific enough. The authors say that "most of the data" was at a resolution of 2hours and of 6 minutes respectively. It is difficult for readers to infer what that means exactly. A table that indicates which stations have fine/coarse resolutions and sparse data would be very helpful. In current manuscript also forces readers infer the magnitude of the distortion that is induced by the different resolutions. How exactly is the coarse scaled data disaggregated? How does the comparison work between aggregated (hence smoothed) fine scale data and disaggregated data? For many readers (including my-self) it will seem strange that a 6 minutes interval for simulating the runoff data is used. The authors claim that the reason for that is to emphasize the main hydrological pro-cess. They do however not explain how works if the input data has a resolution of two hours only.

**4.** The used method need further underpinning. Why are Thiessen Polygons used and not a more sophisticated method (e.g. Kriging)? Especially when the authors conceive that the method can cause problems (Page 9, Lines 14-16). I have similar concerns regarding the use of the genetic algorithm. Why are not more modern and conventional techniques used. In the domain of rainfall-runoff modelling Shuffled Complex Evolution and Dynamically Dimensioned Search come to mind. In general, there also exist more modern versions of genetic/evolutionary algorithms.

**5.** Finally, I would be ready to be convinced otherwise, but I am not sure if the NSE is a well-chosen objective criterion for this sort of study. The low NSE values (that results, according to the authors, from the high resolution) make it difficult to use conventional intuition about the objective measure. Additionally, the chosen events differ largely in from, magnitude, runoff-coefficient and error structure. The amount of evaluated data with low runoffs (before and after the events) also seems to vary. Nonetheless, I believe that in this particular use of the NSE is suboptimal, since information about magnitude

of the events and errors are hidden by its bounded. For this particular case I would therefore propose to use the MSE instead. This would directly express the goodness of fit and the improvements due to the respective calibrations can be seen as relative improvements. Also no additional explanations would be needed to report the jumps of improvements reported at page 9, line 25. Alternatively, it would be useful to provide a sensitivity measure with regard to the individual results.

**6.** Figure 4 needs more explanations and better quality. For some reasons the observation-dots seem not to be equidistant! It is not explained why the model reacts faster than the real system (at least for event 2001 and 2006). At least the strange fit for the event of 2001 seems to be related to this issue, since the "independently calibrated" hydrograph might only be so low because of the large errors at the raising limp. The current resolution of the figure is not good and the depiction is difficult to read. For me it was, for example, not possible to discern what the cluster of observations means at the start of the raising limp of the 2002 event.

—

**Specific Comments:**

S1) This might be nitpicky, but the authors state that the other methods for spatial rainfall uncertainty quantification are not applicable to other basins. Why is that and why does the proposed method not exhibit this problem? The test is only conducted for one basis.

S2) Bootstrapping is a very, very robust method, so the following concern might be of less importance: As far as I know bootstrapping does (still) assume independence of samples. Is this given for rainfall stations? The description of the cluster analysis seems to suggest elsewise. What influence does this have?

S3) **For all Figures**: Please always mention the used basin explicitly in the figure captions. I understand that it is somewhat redundant because the evaluation takes

only place with regard to Quingjian river basins. But, it is very useful for readers who want to get an overview.

S4) **Page 2, Line 9f:** I think rules should be in plural here. Villarini et al. (2008) use a catchment of the size of 125km2, but seem to compare the gauging stations with satellite images with a resolution of 200km2. I therefore believe that the "rules of thumb" should approximate the size of the satellite pixel. I am not sure however, the argument can get finicky here. Lastly, I think one should mention that the study takes place in England.

S.5) **Page 6, Line 28**: What does "adopted" mean in this context? What was changed from the original one. Why is a genetic algorithm used and not a more prevalent method such as the shuffled complex evolution? Why is not a newer evolutionary/genetic algorithm?

S.6) **Page 6, Line 8**: What does it mean that the "influence of topography on rainfall is negligible"?

S.7 ) **Page 7, Line 1-2**: Please specify the "stop criterion" explicitly (number of generations, NSE, ?) How many generations did it take on average?

S8) **Page 9, Line 9**: What kind of independent calibration are we talking about here? Why are parameter interactions not seen as a problem in doing this? Why is only the $K_{ZUS} optimized and not the most sensitive parameter group (e.g. all soil hydraulic conductivity parameters)$?

S9) **Page 9, Lines 25-26**: I would disagree with the statement that the individually calibrated model runs prove that DYRIM is able to represent the rainfall-runoff events in a sufficient way, as long as no evidence is provided that the results are not just due to overfitting (Maybe evaluate the individually derived parameters for the other events to?). I would propose to see the NSE values as hints (if anything). One might also be able to argue that the generated hydrographs can be seen as some sort are upper boundaries or best case scenarios for the DYRIM simulations. Additionally, would it

not be possible to use this information to determine the possible upper bound for the relationship between NSE and measurement side density (Equation 2 and Figure 9 )?

S10) **Page 11, Lines 1-11**: In my opinion this should be part of the method section and not of the results.

S11) **Page 13, Lines 29-31**. Here it is argued that the large errors of the hydrological simulations of the 2006 rainfall event are most likely due the structural and parameter deficiency. Is it possible to plot the cumulated rainfall alongside the cumulate measured runoff for this event (or for all of them)? On basis of the low runoff coefficient and the hydrographs I would (perhaps naively) assume that it there is a bias in the input or the runoff measurements.

—

**Minor Remarks**

Page 1, Line 25: Can you provide additional sources here? Beven, 2001 is a large tome, while Cibin 2014 seems to be focused on ungauged basins (which are not even mentioned as application example).

Page 1, Line 25-29: For me the first sentence of this passage is to intertwined. Could you divide the sentence into two? One point is that hydrologist try to improve the accuracy of simulations and predictions and another point is how this can be achieved (i.e. improving the model structure, better calibration method, better measurements of input data). From there on I think it would be worth to add an additional sentence, arguing why the precipitation is seen important factor for improving simulations and forecasts.

Page 2, Line 1: Please rethink this sentence. The comparison between radar and rainfall station is strangely formulated.

Page 2, Lines 11-13: Please provide additional sources. Four studies are not numerous.

Page2, Line 14: Please recalculate the units from square-miles to square kilometers and use that unit consistently throughout the manuscript.

Page 2, Line 18f. You described what Moulin et al. (2009), but not their results/conclusions. Readers will wonder why not, as they are provided for the previously mentioned studies. Could you expand on that?

Page 3, Lines 19-20: I do not understand what the citations are referring to. Are all these authors proving the advantages of simplicity and high-accuracy?

Page 3, Lines 7: Can somehow you remove the double and in the sentence (maybe use "as well as"? It would improve the readability of the sentence

Page 4, Lines 8-9: I find that argument a bit difficult, in my eyes the bootstrap is a generalization of the jackknife. Maybe compare it to another resampling technique?

Page 4, Line 13: Is it wise to put this argument forward like this. As far as I know, bootstrapping assumes independence of samples. That is not necessarily a property of the population per se, but the sentence could lead to misunderstandings.

Page 4, Line 19-22. You might want to split the sentence to improve readability.

Page 5, Line 13: Change to "are obtained".

Page 5, Line 16: Can you make the following statement more explicit: "..., the bootstrap method is used to traverse most of the combinations of rainfall stations, ..." (emphasis is my own). What does most mean?

Page 5, Line 30f: What exactly does "unit" mean in this context?

Page 6, Line 5: Remove "Then".

Page 6, Line 8: Remove "End".

Page 6, Line 31-31 I do not understand the sentence "This technique promises the parameters independent of the GA and easy to be optimized"

Page 8, Line 1: Write "One rainfall event, which occurred on . . .".

Page 9, Lines 11-12: I do not understand this sentence.

Page 9, Lines 32-24: This statements needs a citation.

Page 10, Lines 3-4: Shouldn't this be part of the results and discussion?

Page 10, Lines 27-31.: Please reformulate.

Page 11, Lines 27-28: Sentence is unclear.

Page 12, Line 10: Please reformulate.

Page 13, Lines 3-4: Write "that even if"

Figure 2: Please rework the plot. The legends are hard to read. It is difficult to grasp the extends of the basins.

Figure 9: Why is the "prediction curve with upper limit" in plot c, higher than the prediction curve without upper limits. The former appears to be fitted for less data. Also, is it possible to provide uncertainty bounds?

---

## Author Comment (AC1) · 9 Oct 2017

**Responses to comments**

We, the authors of the manuscript, appreciate the valuable and constructive comments from Anonymous Referee #1. We will thoroughly revise the manuscript according to these comments. The detailed responses to the comments and questions are as follows.

**General comments**

The authors set out to establish a simple method for estimating the uncertainty of areal rainfall estimates on hydrologic simulations. While such approaches have been broadly considered (and are well cited in the introduction), the authors suggest that existing approaches lack an ability to be extrapolated to other places. Presumably the authors believe their approach addresses this, though that is not explicitly clear to the reviewer. Even if this were the case, unfortunately the modeling study suffers from fatal flaws that prohibit further interpretation of the results. Namely, the authors rely on running a 6-min simulation model with 2-hour forcing (precipitation). This causes multiple problems with model behavior (see specific comments). Additionally, the model clearly suffers from misspecification as a result of overfitting – demonstrating that the curve can be matched but at the expense of process fidelity.

[Answer] We would like to thank the anonymous referee for giving us valuable and constructive comments, which have encouraged us to view our work with much greater insight than before. We will thoroughly revise the manuscript based on these comments. We hope the revision will improve the completeness and accuracy of the results.

Yes, this study aims to propose a general methodology that will not be limited by data, model and river basin. For example, the Qingjian River basin and Longxi River basin are selected as two different cases. With reference to the simulation results, we have to recognize that they are not very satisfying due to several reasons, especially the quality of the rainfall data. We have obtained high-temporal-resolution rainfall data for some of the stations, and we will try to include those new data to get better results in the revised manuscript. Moreover, the overfitting problem will also be carefully investigated in the revision.

Details can be found in the following responses to the specific comments.

**Specific comments**

1. P3.L14-16: Is the argument for this study that a more straightforward methodology is needed? Or simply that this type of study needs to be performed at more sites than it has been to date?

[Answer] The objective of this study is to propose a more straightforward methodology. We will make this clearer in the closing part of the Introduction in the revised manuscript.

2. P3.L24-25: In L12-14 on this page, you argue that results are not reliably extrapolated to other locations, so can your study really accomplish this, as stated?

[Answer] Thanks for the referee's comment. It is worth noting that this study aims at a general method for managing rainfall station density rather than a uniform conclusion on the "representative area of rainfall stations". We believe that the proposed method can be used to extrapolate the results to other locations, e.g., in this study, the analysis of the Longxi River basin is an extension and supplement to that of the upstream Qingjian River basin. This indicates that the proposed method would not be limited by data, model and river basin; however, the results would be. In the revised manuscript, we will make the statement in a less strong tone.

3. P4.L30-31: Why use Thiessen? Why not use something more sophisticated?

[Answer] The calculation of the average basin rainfall needs to be performed hundreds of times in the bootstrap method, which makes the performability as the first factor to be considered. Thiessen polygon method is a classic method and can be easily achieved for batch loops with the ArcGIS. Furthermore, Thiessen polygon method is only used as one representative method to estimate the average basin rainfall, and other methods can also be selected to accomplish this. We can try more interpolation methods in the revision.

4. P6.L1-3: Why are you using 6-min time steps here, when DYRIM performance above (P5.L26-29) is noted as satisfactory for daily to monthly time steps?

[Answer] Actually, the time step used in the DYRIM is determined by the temporal scale of the observed runoff data. For the event-based hydrological simulation of short duration in this study, the minimum time interval of the observed runoff data can be only 6 minutes. That is why we have used 6-min time step rather than the daily or monthly time step here. For hydrological simulations of longer durations, the time step will still be 6-min but the original discharge outputs will be used to calculate the daily or monthly discharge for comparison (e.g., Shi et al., 2015, 2016).

5. P6.L6-8: How many parameters are in DYRIM, how many need to be calibrated?

[Answer] The parameters of DYRIM can be divided into two types (Wang et al., 2015; Shi et al., 2016; Zhang et al., 2016). The first type includes invariant parameters that are used to describe the properties of the land use and soil types, including the field capacity of the topsoil and subsoil layers. The invariant parameters can be determined from field measurements and handbooks and have less influence on the simulated basin runoff. The other type includes all of the sensitive and adjustable parameters, which must be calibrated before model application using the observed rainfall and runoff data. Among them, the most sensitive parameters need to be calibrated are the vertical and lateral saturated conductivities (Table 2). The hillslope runoff-yield model and the main parameters are showed clearly in the following figure. Moreover, the parameters related to the river routing processes such as river manning coefficient should also be concerned.

[Figure]

**Figure. 1** The hillslope runoff-yield model and the main parameters that are used in DYRIM. In this figure, $t$ is time, $W_u$ is the water storage of the topsoil, $Q_{gu}$ is the topsoil drainage, $W_d$ is the water storage of the subsoil and $Q_{gd}$ is the subsoil drainage. $K_{zu}$ is the vertical conductivity of the topsoil layer, which is a function of $K_{zus}$ (the vertical saturated conductivity of the topsoil layer) and $\theta_u(t)$ (the topsoil moisture). $K_{u-d}$ is the vertical conductivity between the topsoil and subsoil, which is a function of $K_{u-ds}$ (the vertical saturated conductivity between the topsoil and subsoil), $\theta_u(t)$ and $\theta_d(t)$ (the subsoil moisture) (From Zhang et al., 2016)

6. P6.L28-29: Why would you use this rather than something like SCE-UA or DDS, which are recognized to be the "best" global optimization algorithms for hydrological applications?

**[Answer]** Thanks for the referee's kind suggestion. Yes, for hydrological model optimization, SCE-UA and DDS are better than GA, and to use SCE-UA (or DDS) may make the results better. However, we have proposed a method for hydrological model calibration which is parallelized with a double-layer structure on HPC systems and it is proved to be valid (Zhang et al., 2016). This method has drawn the attentions of other researchers (e.g., Huang et al., 2016; Kuchar et al., 2016; Nourani and Partoviyan, 2017; Gelleszun et al., 2017). In this study, we just applied this method. It is worth noting that the two layers of parallelization are independent from each other, and then the upper layer is capable of incorporating other optimization algorithms, including SCE-UA (or DDS). Therefore, following the referee's kind suggestion, we may conduct further studies on evaluating the performances of this method by using other methods (e.g., SCE-UA or DDS) in our future work.

7. P9.L3-4: But, if you don't have 6-min forcing data, what does the 6-min runoff simulation get you? I am also concerned by the number of ad hoc adjustments being made in the model implementation (disaggregating rainfall data to smaller time steps, selecting the nearest observed runoff time for NSE evaluation, etc).

**[Answer]** As mentioned in the answer to Comment 4, the time step used in the DYRIM is determined by the temporal scale of the observed runoff data. We have used such fine time step (i.e., 6 minutes) in order to match the exact timing of the observed runoff data, which are instantaneously sampled and have about only

6-minute time steps during the main flooding processes.

Actually, disaggregating sparser rainfall data into smaller time steps (i.e., 2 hours) will have little effect on the precision of hydrological simulation. In contrast, aggregating fine time step rainfall data into larger time steps (i.e., 2 hours) will affect the precision of hydrological simulation since short-duration and high-intensity rains may be homogenised. However, this study focuses on the spatial uncertainty of rainfall rather than the temporal uncertainty, and thus, the measured rainfall data should be pre-treated to have the same time step (i.e., 2 hours).

Moreover, we have obtained high-temporal-resolution rainfall data for some of the stations, and we will try to include those new data to get better simulation results in the revised manuscript. However, how to disaggregate the rainfall data in the stations without new data into the 6-minute time step will be a new problem.

In addition, we believe that selecting the simulated discharge time nearest to the observed runoff time for NSE evaluation is acceptable because fine time step (i.e., 6 minutes) is used in this study and the largest time error will be only 6 minutes. In fact, most of the observed runoff times can be matched by the simulated discharge times; only a very few observed runoff times should be treated as above.

8. P9.L13-14: Exactly, you can't really simulated 6-min runoff with 2-hour rainfall. My question, then, is what value do these results have? The study would be stronger if the data fit the desired framework…
**[Answer]** We fully agree with the referee that this study will be stronger if the input data can fit the desired framework. However, unfortunately, we did not have the measured rainfall data with finer time steps during the preparation of the original manuscript. Now we have obtained high-temporal-resolution rainfall data for some of the stations, and we will try to include those new data to get better simulation results in the revised manuscript. However, how to disaggregate the rainfall data in the stations without new data into the 6-minute time step will be a new problem.

9. P9.L14-16: So why use Thiessen method?!
**[Answer]** We appreciate the referee's comment. Please see the answer to Comment 3 for details.

10. P9.L16-19: This is a clear flaw in the study design. Any interpretation of these results will naturally be influenced by this.
**[Answer]** Yes, we have to recognize that this will influence the simulation results to some extent because short-duration and high-intensity rains may be homogenised. However, such negative impacts are mainly from the aspect of temporal uncertainty rather than spatial uncertainty, and this study aims to estimate the influence of rainfall spatial uncertainty in hydrological simulations. The rainfall temporal uncertainty in hydrological simulations may be further investigated in our future work.

11. P9.L21-22: Why should the vertical saturated conductivity of the topsoil change so much? Shouldn't this value theoretically be constant? Aren't these changes

representative of compensation of other errors in the modeling?

**[Answer]** Thanks for the constructive comment. The vertical saturated conductivity of the topsoil layer ($K_{zus}$) controls the surface infiltration rate and primarily influences the infiltration-excess runoff. For example, surface roughness and vegetation of a hillslope affect the residence time and infiltration rate of water on its surface, and finally influence the total amount of infiltration. Yes, the vertical saturated conductivity of the topsoil should not change so much for a river basin. In this study, we first calibrate this parameter based on the observed data of all the three selected events and find that the simulation results are not so good. Therefore, we further calibrate this parameter independently for the three selected events in order to demonstrate the potential performance of the DYRIM. You may see that the simulation results using the same vertical saturated conductivity of the topsoil for the three selected events are also listed in Table 1.

12. P9.L25-26: But it doesn't prove that it achieves this performance for an appropriate reason. This is just curve fitting, isn't it?

**[Answer]** Yes, we agree that it is curve fitting, which is a way for evaluating the performance of the model. The results show that the topsoil vertical saturated conductivity $K_{zus}$ is sensitive in this case, and using the same $K_{zus}$ value may make the model performance unsatisfactory.

13. P10.L6: How would you utilize these results to improve rainfall gauging density or placement? Can you predict these locations without already having measured rainfall? If not, then what is the benefit?

**[Answer]** We appreciate the referee's insightful comment. We have to recognize that locations of rainfall stations may not be predicted or determined from this method without the measured rainfall data. However, this study aims at a general method for managing rainfall station density, which would not be limited by data, model and river basin. Moreover, the proposed method can give the suggestion of the proper $D_S$ in a river basin using the proposed fitting of Equation (2). For example, this study indicates that the controlling area of each rainfall station should be about 40 km$^2$ for hydrological simulation at the hourly time scale in the middle Yellow River basin (see Section 4.5). However, a uniform conclusion on the "representative area of rainfall stations" cannot be obtained by using this method because the results would be influenced by rainfall patterns in different river basins.

**Technical corrections**

1. P2.1-2: "more" reliable…

**[Answer]** We will revise this in the revision. Thanks.

2. P3.3: Change "good fit" to "goodness of fit"

**[Answer]** We will revise this in the revision. Thanks.

**References:**

Gelleszun, M., Kreye, P., Meon, G., 2017. Representative parameter estimation for hydrological models using a lexicographic calibration strategy. Journal of Hydrology, doi: 10.1016/j.jhydrol.2017.08.015.

Huang, P.N., Li, Z.J., Chen, J., Li, Q.L., Yao, C., 2016. Event-based hydrological modeling for detecting dominant hydrological process and suitable model strategy for semi-arid catchments. Journal of Hydrology, 542, 292-303.

Kuchar, S., Podhoranyi, M., Vavrik, R., Portero, A., 2016. Dynamic computing resource allocation in online flood monitoring and prediction. IOP Conference Series: Earth and Environmental Science, 39, 012061.

Nourani, V., Partoviyan, A., 2017. Hybrid denoising-jittering data pre-processing approach to enhance multi-step-ahead rainfall–runoff modeling. Stochastic Environmental Research and Risk Assessment, doi: 10.1007/s00477-017-1400-5.

Shi, H.Y., Li, T.J., Liu, R.H., Chen, J., Li, J.Y., Zhang, A., Wang, G.Q., 2015. A service-oriented architecture for ensemble flood forecast from numerical weather prediction. Journal of Hydrology, 527, 933-942.

Shi, H.Y., Li, T.J., Wang, K., Zhang, A., Wang, G.Q., Fu, X.D., 2016. Physically-based simulation of the streamflow decrease caused by sediment-trapping dams in the middle Yellow River. Hydrological Processes, 30(5), 783-794.

Wang, G.Q., Fu, X.D., Shi, H.Y., Li, T.J., 2015. Watershed sediment dynamics and modeling: a watershed modeling system for Yellow River. In Yang C.T. and Wang L.K. (eds), Advances in Water Resources Engineering, Handbook of Environmental Engineering, Volume 14, Chapter 1, 1-40. Springer Cham Heidelberg New York Dordrecht London.

Zhang, A., Li, T.J., Si, Y., Liu, R.H., Shi, H.Y., Li, X., Li, J.Y., Wu, X., 2016. Double-layer parallelization for hydrological model calibration on HPC systems. Journal of Hydrology, 535, 737-747.

---

## Author Comment (AC2) · 9 Oct 2017

**Responses to comments**

We, the authors of the manuscript, appreciate the valuable and constructive comments from the Referee, D. Klotz. We will thoroughly revise the manuscript according to these comments. The detailed responses to the comments and questions are as follows.

**General comments**

The present study proposes a bootstrap based quantification of the spatial rainfall uncertainty for rainfall-runoff modelling. The main body of the work uses aforementioned bootstrap method is used to select between different numbers of stations and different groups of stations. The influence of the number and grouping is then quantified via the Nash Sutcliffe Efficiency and exponential fit is used to derive a heuristic for the estimation of the optimal densities of rainfall measurement stations. The whole analysis is based on three manually selected rainfall events from 2001, 2002 and 2006, with data from of the Qingjian basin. Additionally, rainfall data from a second basin (Longxi) is used to provide a basic comparison about the scale and uniqueness of the conclusions.

The topic as such is certainly relevant for the study of the rainfall-runoff relationship as such and would fit a HESS. I do also conceive that the endeavor of the authors poses major challenges. Rainfall process can be extremely heterogeneous and as the authors state a quantification of the uncertainties depends multiple interplaying phenomena (such as the position of the meteorological stations, the unique futures of the studied events or the elevation profiles of a catchment). So, the authors did probably a good job, if we consider the possible difficulties that are inherent in such a study. I am nevertheless left with many concerns. The most important wants are listed in the following points, minor comments and questions are given later.

**[Answer]** We would like to thank the referee for giving us valuable and constructive comments, which have encouraged us to view our work with much greater insight than before. We will thoroughly revise the manuscript based on these comments. We hope the revision will improve the completeness and accuracy of the results. Please see our responses to all the comments in the following for details.

1. First and foremost, I believe that considerable attention should be given to improve the introduction in special and the readability of the manuscript in general. The whole manuscript needs a clearer structure, the aims must be defined in a clearer fashion (maybe less focus on different aspects and more focus on the main topics) and the manuscript as a whole should meander less between topics. Arguments about the computational expense, for example, can be found throughout the script, but no chapter is devoted to it.

**[Answer]** We appreciate the referee's constructive comment. In the revision, we will improve the introduction part through citing more related papers and deleting the

irrelevant ones. We will also reorganize the text with a clearer structure and further highlight the main topics of this manuscript. As suggested by the referee, we will add some sentences to discuss the computational expense of this method in the revised manuscript.

2. In reference to my last point, I have to say that I truly believe that is worth to discuss the computational expense of the method as such. Currently hints about the computational efficacy are spread throughout the manuscript, but I miss a dedicated chapter and quantitative arguments. The method seems quite computationally expensive as the hydrological model needs to be executed approximately N*K times. Now what I would like to know is what it does mean quantitatively. How long does a run of the used r-r model take, and how long does an execution of the bootstrap need. Especially the last point provides a plethora of possible topics/experiment: Influence of shorter and longer time series, usage of the bootstrap without and with parallelization, usage of the model with and without parallelization, on a cluster or a normal pc, etc.
**[Answer]** The selected upper Qingjian River basin has a moderate size. The number of hillslope-channel unit in this river basin is about 8000, and a serial simulation using the DYRIM costs about 5 minutes. The selected basin is only for the demonstration of the proposed methods in this manuscript. Further application of the methods to larger river basins and to calibrate more parameters will of course cost more computational time, and the proposed methods will keep effective.

   The model simulation was implemented on the Microsoft Windows Azure cloud computing platform, allocated with a total number of 80 processor cores. With the Windows Azure configuration, the simulation time of the DYRIM with 4 processor cores for one event in the upper Qingjian River basin was 130 s.

3. The actual time-resolution of the modelling needs some further discussion and explanations. I think the main problem here is that the descriptions are not specific enough. The authors say that "most of the data" was at a resolution of 2 hours and of 6 minutes respectively. It is difficult for readers to infer what that means exactly. A table that indicates which stations have fine/coarse resolutions and sparse data would be very helpful. In current manuscript also forces readers infer the magnitude of the distortion that is induced by the different resolutions. How exactly is the coarse scaled data disaggregated? How does the comparison work between aggregated (hence smoothed) fine scale data and disaggregated data? For many readers (including myself) it will seem strange that a 6 minutes interval for simulating the runoff data is used. The authors claim that the reason for that is to emphasize the main hydrological process. They do however not explain how works if the input data has a resolution of two hours only.
[Answer] Thanks for the useful comment. Actually, the time step used in the DYRIM is determined by the temporal scale of the observed runoff data. For the event-based hydrological simulation in this study, the minimum time interval of the observed runoff data can be only 6 minutes, and therefore, we have used 6-min time step rather

than the larger time step. We have to select the simulated discharge time nearest to the observed runoff time for the NSE evaluation in order to reduce the time error. In that case, most of the observed runoff times can be matched by the simulated discharge times; only a very few observed runoff times should be treated as above.

Actually, disaggregating sparser rainfall data into smaller time steps (i.e., 2 hours) will have little effect on the precision of hydrological simulation. In contrast, aggregating fine time step rainfall data into larger time steps will affect the precision of hydrological simulation since short-duration and high-intensity rains may be homogenised. However, this study focuses on the spatial uncertainty of rainfall rather than the temporal uncertainty, and thus, the measured rainfall data should be pre-treated to have the same time step (i.e., 2 hours).

We fully agree with the referee that this study will be stronger if the input data can fit the desired framework. However, unfortunately, we did not have the measured rainfall data with finer time steps during the preparation of the original manuscript. Most (up to 95%) of the time steps of the measured rainfall data are 2 hours, and the measured rainfall data with finer time steps are archived by the local hydrology bureau with some confidentiality. This is the main reason why we have to adjuste the time steps of all the measured rainfall data to 2 hours. Now, we have obtained high-temporal-resolution rainfall data for some of the stations, and we will try to include those new data to get better simulation results in the revised manuscript.

4. The used method need further underpinning. Why are Thiessen Polygons used and not a more sophisticated method (e.g. Kriging)? Especially when the authors conceive that the method can cause problems (Page 9, Lines 14-16). I have similar concerns regarding the use of the genetic algorithm. Why are not more modern and conventional techniques used. In the domain of rainfall-runoff modelling Shuffled Complex Evolution and Dynamically Dimensioned Search come to mind. In general, there also exist more modern versions of genetic/evolutionary algorithms.

[Answer] We appreciate the referee's comment. The calculation of the average basin rainfall needs to be performed hundreds of times in the bootstrap method so that the performability should be regarded as the first consideration. Thiessen polygon method can be easily achieved for batch loops with the ArcGIS, and it is worth noting that Thiessen polygon method is only adopted as one representative way to estimate the average basin rainfall, which can also be realized by using other methods.

In this manuscript, GA is selected as the example optimization algorithm that employed in the double-layer parallelization (Zhang et al., 2016), mainly because GA is simple and widely-accepted. For hydrological model optimization, SCE-UA and DDS are considered better than GA, and the double-layer parallelization plus SCE-UA (or DDS) will be better than plus GA. However, the two layers of parallelization are independent from each other, and then the upper layer is capable of incorporating other optimization algorithms, including the SCE-UA (or DDS). Therefore, the adoption of GA will not largely influence the contribution of this manuscript on hydrological uncertainty.

5. Finally, I would be ready to be convinced otherwise, but I am not sure if the NSE is a well-chosen objective criterion for this sort of study. The low NSE values (that results, according to the authors, from the high resolution) make it difficult to use conventional intuition about the objective measure. Additionally, the chosen events differ largely in from, magnitude, runoff-coefficient and error structure. The amount of evaluated data with low runoffs (before and after the events) also seems to vary. Nonetheless, I believe that in this particular use of the NSE is suboptimal, since information about magnitude of the events and errors are hidden by its bounded. For this particular case I would therefore propose to use the MSE instead. This would directly express the goodness of fit and the improvements due to the respective calibrations can be seen as relative improvements. Also no additional explanations would be needed to report the jumps of improvements reported at page 9, line 25. Alternatively, it would be useful to provide a sensitivity measure with regard to the individual results.

**[Answer]** The Nash-Sutcliffe coefficient of efficiency (NSE) has been used as one of the very popular indices to assess the performance of hydrological models, especially for the event-based simulation. The peak time and peak value can be well evaluated by using the NSE. However, we may also use the MSE as another objective criterion and compare the results against those obtained by using the NSE. Moreover, the three events with different rainfall-runoff responses were selected in this study because they are partially caused by different spatial rainfall patterns, and they cannot be accurately simulated by using the same set of parameters. That is why we have tried to calibrate the parameters for these three events independently.

6. Figure 4 needs more explanations and better quality. For some reasons the observation-dots seem not to be equidistant! It is not explained why the model reacts faster than the real system (at least for event 2001 and 2006). At least the strange fit for the event of 2001 seems to be related to this issue, since the "independently calibrated" hydrograph might only be so low because of the large errors at the raising limp. The current resolution of the figure is not good and the depiction is difficult to read. For me it was, for example, not possible to discern what the cluster of observations means at the start of the raising limp of the 2002 event.

**[Answer]** Thanks for the comment. In this study, the minimum time interval of the observed runoff data is only 6 minutes during the flood, while the time intervals are much larger before or after the flood. That is why the observation-dots are not equidistant.

For the precision of the simulation results, with the scale ranging from monthly to daily, former studies of the DYRIM have achieved pretty good results and the NSE could easily reach a satisfactory value, for example, higher than 0.85 (Wang et al., 2015). What we are endeavoring to do in this manuscript is to achieve the simulation of flood events with an hourly scale. In the DYRIM, the temporal resolution of runoff simulation is 6 minutes, but each rainfall data point with a 2-hour step is uniformly assigned to corresponding simulation time steps. The time step of rainfall may have a considerable impact on the results, resulting in the flatter shape of simulated flow

discharges processes (Fig. 4). To better simulate such short-duration high-intensity rainfall-runoff events, rainfall observations with higher spatial and temporal resolutions are badly needed. When finer rainfall records are obtained from concurrent meteorological stations and satellite remote sensing, e.g., with a time step as fine as 1 minute and a spatial resolution at 8 km, the time step of hydrological models will be even shorter than 6 minutes, and the hillslope-channel units will be smaller than 0.1 km$^2$. In such a situation, the computational consumption of hydrological model calibration will increase dramatically, and the proposed double layer parallelism will be one of the necessities.

**Specific comments**

S1) This might be nitpicky, but the authors state that the other methods for spatial rainfall uncertainty quantification are not applicable to other basins. Why is that and why does the proposed method not exhibit this problem? The test is only conducted for one basis.

**[Answer]** Former studies mainly focused on the certain river basins or rainfall events with specific method such as the conditional simulation, and this may lead to different results of spatial rainfall variability, such as the rules proposed by Villarini et al. (2008) (e.g., more than 15 stations over an area of approximately 135 km$^2$ were necessary to estimate the true areal rainfall at a three-hour scale with an error less than 20%).

To the best of our knowledge, this paper is an innovative work to propose such a bootstrap method for uncertainty estimation. This contribution primarily aims at the construction and realization of the method architecture, rather than an uncertainty case study. Furthermore, the hydrological model simulation and the bootstrap method proposed in this paper are independent from each other. The capability of the proposed system may incorporate different hydrological models and other techniques beyond the GA and Thiessen Polygon. This is also an important advantage of the proposed approach.

S2) Bootstrapping is a very, very robust method, so the following concern might be of less importance: As far as I know bootstrapping does (still) assume independence of samples. Is this given for rainfall stations? The description of the cluster analysis seems to suggest elsewise. What influence does this have?

**[Answer]** Data series from any rainfall station are recorded independently at the point where the rainfall station locates, and data from all rainfall stations are indispensable for providing areal rainfall estimates. What we have done for cluster analysis is positive that a certain rainfall in an area has the inherent connection because of the weather condition and physical climate effects. What we want to do is to achieve the more accurate simulation results and complete understanding of the rainfall spatial uncertainty by using the cluster analysis and the bootstrap method based on the available data.

S3) For all Figures: Please always mention the used basin explicitly in the figure

captions. I understand that it is somewhat redundant because the evaluation takes only place with regard to Qingjian river basins. But, it is very useful for readers who want to get an overview.

**[Answer]** We will revise this in the revision. Thanks.

S4) Page 2, Line 9f: I think rules should be in plural here. Villarini et al. (2008) use a catchment of the size of 125km$^2$, but seem to compare the gauging stations with satellite images with a resolution of 200km$^2$. I therefore believe that the "rules of thumb" should approximate the size of the satellite pixel. I am not sure however, the argument can get finicky here. Lastly, I think one should mention that the study takes place in England.

**[Answer]** We will revise this in the revision. Thanks.

S.5) Page 6, Line 28: What does "adopted" mean in this context? What was changed from the original one. Why is a genetic algorithm used and not a more prevalent method such as the shuffled complex evolution? Why is not a newer evolutionary/genetic algorithm?

**[Answer]** We are sorry for the misleading in this part. The genetic algorithm (GA) was firstly proposed by Holland (1975), but the GA implementation using binary and real coded variables with the newer C++ version that we adopted in this study was developed by Deb (1997). This implementation employed in this paper treats the parameters to be optimized as real numbers with simulated binary crossover and real-parameter mutation. This technique promises the hydrological model parameters independent of the GA. We will further revise this part in the revision.

For the reason of choosing the GA rather than other methods, please refer to our response to Point 4 of the General comments for details.

S.6) Page 6, Line 8: What does it mean that the "influence of topography on rainfall is negligible"?

**[Answer]** As is known, the topography has impacts on the rainfall, particularly for the mountainous regions. It should be taken into account in the hydrological simulation. However, this kind of influence is not a major problem in this study and has not been considered. Nevertheless, we will remove this statement in the revision.

S.7) Page 7, Line 1-2: Please specify the "stop criterion" explicitly (number of generations, NSE, ?) How many generations did it take on average?

**[Answer]** During the fitness evaluation of the GA, a new generation of the model parameter is proposed to explore more parameter combinations until the stop criterion or the maximum number of generations is reached. We want to explore the fluctuation range of the NSE in uncertainty estimation, so we set the maximum number of generations (e.g., 10 generations in this study) as the stop criterion.

S8) Page 9, Line 9: What kind of independent calibration are we talking about here? Why are parameter interactions not seen as a problem in doing this? Why is only the

$K_{ZUS}$ optimized and not the most sensitive parameter group (e.g. all soil hydraulic conductivity parameters)?

[Answer] There are two kinds of schemes for the model calibration in this study. One is the independent calibration on each rainfall-runoff event using different sets of parameter combinations, and the other is the calibration on all the events sharing the same set of parameter combination. When the model is calibrated on the events independently one by one, more precise results would be obtained to show the potential performance of the model. However, in such a scheme, the optimized parameters will adapt to each described distribution of rainfall by the stations. Therefore, to insulate the effect of rainfall spatial distribution from model parameters, at least in an average sense, the three events should be considered comprehensively through using the same set of parameter combination.

S9) Page 9, Lines 25-26: I would disagree with the statement that the individually calibrated model runs prove that DYRIM is able to represent the rainfall-runoff events in a sufficient way, as long as no evidence is provided that the results are not just due to overfitting (Maybe evaluate the individually derived parameters for the other events to?). I would propose to see the NSE values as hints (if anything). One might also be able to argue that the generated hydrographs can be seen as some sorts are upper boundaries or best case scenarios for the DYRIM simulations. Additionally, would it not be possible to use this information to determine the possible upper bound for the relationship between NSE and measurement side density (Equation 2 and Figure 9)?

[Answer] We have to recognize that the model performances are not so good for certain events and that is why we have conducted the independent calibration on each rainfall-runoff event using different sets of parameter combinations. You may regard the latter results as the upper boundaries or the best scenarios for the simulations of these events. However, we are sorry to say that this information has not been used to determine the possible upper bounds for the relationships presented in Equation 2 and Figure 9 yet. We may conduct further study on this in our future work.

S10) Page 11, Lines 1-11: In my opinion this should be part of the method section and not of the results.

[Answer] We will add a new subsection (i.e., 3.3 Uncertainty quantization of basin rainfall and simulated runoff) and move this part to that subsection. Thanks.

S11) Page 13, Lines 29-31. Here it is argued that the large errors of the hydrological simulations of the 2006 rainfall event are most likely due the structural and parameter deficiency. Is it possible to plot the cumulated rainfall alongside the cumulate measured runoff for this event (or for all of them)? On basis of the low runoff coefficient and the hydrographs I would (perhaps naively) assume that it there is a bias in the input or the runoff measurements.

[Answer] Thank for the insightful suggestion. A new figure comparing the cumulated rainfall and measured runoff will be plotted in the revised manuscript.

**Minor Remarks**

Page 1, Line 25: Can you provide additional sources here? Beven, 2001 is a large tome, while Cibin 2014 seems to be focused on ungauged basins (which are not even mentioned as application example).
**[Answer]** We will cite more papers (e.g., Singh and Prevert, 2002; Fares et al., 2014) in the revision. Thanks.

Page 1, Line 25-29: For me the first sentence of this passage is to intertwined. Could you divide the sentence into two? One point is that hydrologist try to improve the accuracy of simulations and predictions and another point is how this can be achieved (i.e. improving the model structure, better calibration method, better measurements of input data). From there on I think it would be worth to add an additional sentence, arguing why the precipitation is seen important factor for improving simulations and forecasts.
**[Answer]** In the revision, we will divide this sentence into two and also add an additional sentence to argue why precipitation is seen important factor for improving simulations and forecasts. Thanks.

Page 2, Line 1: Please rethink this sentence. The comparison between radar and rainfall station is strangely formulated.
**[Answer]** We will revise this in the revision. Thanks.

Page 2, Lines 11-13: Please provide additional sources. Four studies are not numerous.
**[Answer]** We will provide more details and cite more papers (e.g., Bedient et al., 2000; Smith et al., 2004; Casper et al., 2009; He et al., 2013) in the revision. Thanks.

Page2, Line 14: Please recalculate the units from square-miles to square kilometers and use that unit consistently throughout the manuscript.
**[Answer]** We will revise this in the revision. Thanks.

Page 2, Line 18f. You described what Moulin et al. (2009), but not their results/conclusions. Readers will wonder why not, as they are provided for the previously mentioned studies. Could you expand on that?
**[Answer]** We will provide more details about the study of Moulin et al. (2009) in the revision. Thanks.

Page 3, Lines 19-20: I do not understand what the citations are referring to. Are all these authors proving the advantages of simplicity and high-accuracy?
**[Answer]** The citations here are referred to the analysis or application of the bootstrap method in the hydrological model, and several of them have proved the advantages (e.g., simplicity and high-accuracy). We will further revise this sentence to make it

clearer. Thanks.

Page 3, Lines 7: Can somehow you remove the double and in the sentence (maybe use "as well as"? It would improve the readability of the sentence
**[Answer]** We will use "as well as" instead of "and" in the revision. Thanks.

Page 4, Lines 8-9: I find that argument a bit difficult, in my eyes the bootstrap is a generalization of the jackknife. Maybe compare it to another resampling technique?
**[Answer]** The jackknife is thought of as a linear expansion method for approximating the bootstrap. Bootstrap methods are more widely applicable than the jackknife and also more dependable. We will further revise this sentence to make it clearer. Thanks.

Page 4, Line 13: Is it wise to put this argument forward like this. As far as I know, bootstrapping assumes independence of samples. That is not necessarily a property of the population per se, but the sentence could lead to misunderstandings.
**[Answer]** We will remove this sentence in the revision. Thanks.

Page 4, Line 19-22. You might want to split the sentence to improve readability.
**[Answer]** We will revise this sentence in the revision. Thanks.

Page 5, Line 13: Change to "are obtained".
**[Answer]** We will revise this in the revision. Thanks.

Page 5, Line 16: Can you make the following statement more explicit: "…, the bootstrap method is used to traverse most of the combinations of rainfall stations, …" (emphasis is my own). What does most mean?
**[Answer]** In Section 2.1, we have explained the repetition of runoff simulation and estimation for the average basin rainfall by using the bootstrap method, respectively. "*To reduce the repetition of simulations, for a certain combination, a certain number of stations in a certain group are randomly selected only once*." The loop traversed all the combinations of station groups in rainfall estimation, but a few combinations in runoff simulation were skipped based on the station classification.

Page 5, Line 30f: What exactly does "unit" mean in this context?
**[Answer]** The "unit" means the basic hillslope-channel unit. This unit is an independent rainfall runoff module since the DYRIM is a penalization hydrological model. We will further revise this sentence to make it clearer. Thanks.

Page 6, Line 5: Remove "Then".
**[Answer]** We will remove "Then" in the revision. Thanks.

Page 6, Line 8: Remove "End".
**[Answer]** This sentence will be removed in the revision. Thanks.

Page 6, Line 31-31 I do not understand the sentence "This technique promises the parameters independent of the GA and easy to be optimized"
**[Answer]** The parameters from the hydrological model are independent of the GA, which means these parameter combinations are input and output of the optimization programs. We will further revise this sentence to make it clearer. Thanks.

Page 8, Line 1: Write "One rainfall event, which occurred on …".
**[Answer]** We will revise this in the revision. Thanks.

Page 9, Lines 11-12: I do not understand this sentence.
**[Answer]** The main reason of poor calibration results is the hourly resolution of runoff results using rainfall input with lower temporal resolution. Better precision of simulation can be found in daily to monthly results, with satisfactory NSEs higher than 0.8 (Shi et al., 2016; Wang et al., 2015). In order to demonstrate the potential performance of the DYRIM, the topsoil vertical saturated conductivity $K_{zus}$ was calibrated on the three events independently one by one (as explained in the following paragraph).

Page 9, Lines 32-24: This statement needs a citation.
**[Answer]** We will cite one paper (Huang et al., 2016) in the revision. Thanks.

Page 10, Lines 3-4: Shouldn't this be part of the results and discussion?
**[Answer]** We will move this sentence to the Section 5. Thanks.

Page 10, Lines 27-31.: Please reformulate.
**[Answer]** We will revise this in the revision. Thanks.

Page 11, Lines 27-28: Sentence is unclear.
**[Answer]** We will provide some explanation in the revision. Thanks.

Page 12, Line 10: Please reformulate.
**[Answer]** We will revise this in the revision. Thanks.

Page 13, Lines 3-4: Write "that even if"
**[Answer]** We will revise this in the revision. Thanks.

Figure 2: Please rework the plot. The legends are hard to read. It is difficult to grasp the extends of the basins.
**[Answer]** The legends of this figure will be reworked. Thanks.

Figure 9: Why is the "prediction curve with upper limit" in plot c, higher than the prediction curve without upper limits. The former appears to be fitted for less data. Also, is it possible to provide uncertainty bounds?
**[Answer]** In the third case, the NSE value was low, and the higher $D_S$ values (i.e., 6 to

11) were selected for curve fitting in Figure 9c (the dash lines). We found that the low NSE value of the third case might be caused by the hydrological model structure and parameters rather than the spatial uncertainty of rainfall records. However, in terms of the other two cases, the proposed fitting of Equation (2) would be capable to reveal the possible improvement of simulation. Moreover, the uncertainty analysis for rainfall input combined with model structure and parameters could provide more complete and accurate uncertainty bounds.

**References:**

Bedient, P.B., Hoblit, B.C., Gladwell, D.C., Vieux B.E., 2000. NEXRAD radar for flood prediction in Houston. Journal of Hydrologic Engineering, 5(3), 269-277.

Casper, M.C., Herbst, M., Grundmann, J., Buchholz, O., Bliefernicht, J., 2009. Influence of rainfall variability on the simulation of extreme runoff in small catchments. Hydrologie Und Wasserbewirtschaftung, 53(3), 134-139.

Deb, K., 1997. Genetic algorithm in search and optimization: the technique and applications. Proc.of Int.workshop on Soft Computing & Intelligent Systems, 58-87.

Fares, A., Awal, R., Michaud, J., Chu, P.S., Fares, S., Kodama, K., Rosener, M., 2014. Rainfall-runoff modeling in a flashy tropical watershed using the distributed HL-RDHM model. Journal of Hydrology, 519, 3436-3447.

He, X., Sonnenborg, T.O., Refsgaard, J.C., Vejen, F., Jensen, K.H., 2013. Evaluation of the value of radar QPE data and rain gauge data for hydrological modeling. Water Resources Research, 49(9), 5989-6005.

Huang, P.N., Li, Z.J., Chen, J., Li, Q.L., Yao, C., 2016. Event-based hydrological modeling for detecting dominant hydrological process and suitable model strategy for semi-arid catchments. Journal of Hydrology, 542, 292-303.

Shi, H.Y., Li, T.J., Wang, K., Zhang, A., Wang, G.Q., Fu, X.D., 2016. Physically-based simulation of the streamflow decrease caused by sediment-trapping dams in the middle Yellow River. Hydrological Processes, 30(5), 783-794.

Singh, V.P, Prevert, D.K., 2002. Mathematical models of large watershed hydrology. Water Resources: Highlands Ranch, Colo., 891.

Smith, M.B., Koren, V.I., Zhang, Z., Reed, S.M., Pan, J.J., Moreda, F., 2004. Runoff response to spatial variability in precipitation: an analysis of observed data. Journal of Hydrology, 298(1), 267-286.

Wang, G.Q., Fu, X.D., Shi, H.Y., Li, T.J., 2015. Watershed sediment dynamics and modeling: a watershed modeling system for Yellow River. In Yang C.T. and Wang L.K. (eds), Advances in Water Resources Engineering, Handbook of Environmental Engineering, Volume 14, Chapter 1, 1-40. Springer Cham Heidelberg New York Dordrecht London.

Zhang, A., Li, T.J., Si, Y., Liu, R.H., Shi, H.Y., Li, X., Li, J.Y., Wu, X., 2016. Double-layer parallelization for hydrological model calibration on HPC systems. Journal of Hydrology, 535, 737-747.